# ALIGNING RELATIONAL LEARNING WITH LIPSCHITZ FAIRNESS

**Yaning Jia**[†]**, Chunhui Zhang**[†]**, Soroush Vosoughi**[*]
Dartmouth College, Hanover, NH, USA
HUST, Hubei, China

## ABSTRACT

Relational learning has gained significant attention, led by the expressiveness of Graph Neural Networks (GNNs) on graph data. While the inherent biases in common graph data are involved in GNN training, it poses a serious challenge to constraining the GNN output perturbations induced by input biases, thereby safeguarding fairness during training. The Lipschitz bound, a technique from robust statistics, can limit the maximum changes in the output concerning the input, taking into account associated irrelevant biased factors. It is an efficient and provable method to examine the output stability of machine learning models without incurring additional computational costs. Recently, its use in controlling the stability of Euclidean neural networks, the calculation of the precise Lipschitz bound remains elusive for non-Euclidean neural networks like GNNs, especially within fairness contexts. However, no existing research has investigated Lipschitz bounds to shed light on stabilizing the GNN outputs, especially when working on graph data with implicit biases. To narrow this gap, we begin with the general GNNs operating on relational data, and formulate a Lipschitz bound to limit the changes in the output regarding biases associated with the input. Additionally, we theoretically analyze how the Lipschitz bound of a GNN model could constrain the output perturbations induced by biases learned from data for fairness training. We experimentally validate the Lipschitz bound's effectiveness in limiting biases of the model output. Finally, from a training dynamics perspective, we demonstrate why the theoretical Lipschitz bound can effectively guide the GNN training to better trade-off between accuracy and fairness.

## 1 INTRODUCTION

Relational learning on network data has become ubiquitous in a range of real-world applications, such as social media (Vosoughi et al., 2016; Shalaby et al., 2017; Huang et al., 2021a; Li et al., 2021; Tian et al., 2023a; Ouyang et al., 2024), drug discovery (Takigawa & Mamitsuka, 2013; Li et al., 2017), and knowledge engineering (Rizun, 2019; Wang et al., 2018). This surge in the use of graph data has catalyzed advances in learning algorithms that incorporate Graph Neural Networks (GNNs) with deep learning techniques (Gori et al., 2005; Scarselli et al., 2005; Li et al., 2016; Hamilton et al., 2017; Xu et al., 2019; Zhang et al., 2022, 2023a,b,c; Liu et al., 2023; Yuan et al., 2024). Among GNN architectures, Graph Convolutional Networks (GCNs) (Kipf & Welling, 2017; Zhang & Chen, 2018; Fan et al., 2019) stand out due to their use of convolutional layers and message-passing mechanisms to enable effective learning on graphs.

Amidst the widespread success of GNNs in diverse applications, there has been growing societal concern for developing ethical and prosocial learning algorithms for networks (Vosoughi et al., 2017, 2018; Dong et al., 2021; Lahoti et al., 2019; Kang et al., 2020; Mujkanovic et al., 2022). Many existing approaches, however, fall short of explicating the interactions between the GNN model and its graph-based training data. This opacity hampers the model's parameter tunability, failing to adequately address latent biases in the graph data, thereby undermining model reliability post-training. In this context, our objective is to deepen our understanding of the input-output dynamics of GNNs

---

[*]Corresponding to `soroush.vosoughi@dartmouth.edu`. [†] YJ and CZ contributed equally and are listed alphabetically by last name. YJ, from HUST, was mentored by CZ and SV from Dartmouth College.

in relation to model parameters. Specifically, we confront the following pivotal question: *How can we minimize unintended shifts in GNN output, especially when the graph training data encompasses implicit, learnable biases, without resorting to computationally intensive methods?*

This question is crucial in scenarios where the training graph data may contain spurious statistics that result in unfair or inappropriate correlations. Our work aims to provide a mechanism that curtails such fluctuations in the GNN model's predictions during training, ensuring they are not disproportionately influenced by unfair factors. The implications of this work extend to enhancing GNN generalization, debiasing GNNs, and safeguarding against data perturbations by regularizing output changes. Although existing GNN training methodologies have yielded impressive results, they often lack interpretability, particularly in understanding how layer-by-layer learned biases affect output stability or fairness (Dong et al., 2021; Kang et al., 2020; Liao et al., 2021; Li et al., 2021; Yue et al., 2022).

In robust statistics, Lipschitz bounds serve as valuable tools for assessing maximal output changes in response to input biases (Virmaux & Scaman, 2018; Fazlyab et al., 2019; Jordan & Dimakis, 2020; Latorre et al., 2020; Huang et al., 2021b). Prior work (Dwork et al., 2012; Shi et al., 2022; Agarwal et al., 2021) has mainly focused on fairness and robustness in MLPs, CNN, or statistical models, often relying on explicitly labeled sensitive attributes, such as gender, race, and age.

Our research distinguishes itself by extending fairness considerations specifically to the realm of relational learning on graphs. We achieve this by constructing a fairness framework that does not necessitate manual annotation of sensitive attributes. Instead, our approach is designed to identify and mitigate biases inherent in node features and connectivity patterns within real-world graph data. Our methodology aligns closely with the principles of ranking-based individual fairness as articulated by Dong et al. (2021). In this framework, fairness is guaranteed if the relative ordering of instances, characterized by ranking lists based on the similarity between instance $i$ and other instances, remains consistent in both input and output spaces. Importantly, this ordering should not be distorted by irrelevant or hidden biases in the input data. When these conditions are met, the model output will inherently satisfy individual fairness criteria. To operationalize these principles, we introduce a computational strategy that leverages the Jacobian matrix for the efficient calculation of the Lipschitz bound. Our computational approach utilizes intermediate tensors – specifically, the gradients of the model's outputs with respect to its inputs – in PyTorch, facilitating practical and scalable implementations. This is particularly advantageous for large-scale graph datasets, an area where existing methods often falter in terms of scalability. Our contributions can be summarized as:

- We introduce a Lipschitz bound specifically tailored for GNNs to align with rank-based individual fairness. This allows us to characterize and constrain the perturbations in GNN predictions induced by input biases.

- We develop an efficient method for computing the Lipschitz bound of GNNs by leveraging the Jacobian matrix, thereby accommodating the complex topology of graph structures for practical implementation.

- Through theoretical and empirical validation, we demonstrate that our approach is both versatile and effective, serving as a plug-and-play solution that can enhance existing fairness-oriented graph learning methods.

## 2 PRELIMINARIES

**Lipschitz Bound** A function $f : \mathbb{R}^n \to \mathbb{R}^m$ is said to be Lipschitz continuous on an input set $\mathcal{X} \subseteq \mathbb{R}^n$ if there exists a bound $K \geq 0$ such that for all $\boldsymbol{x}, \boldsymbol{y} \in \mathcal{X}$, $f$ satisfies:

$$\|f(\boldsymbol{x}) - f(\boldsymbol{y})\| \leq K \|\boldsymbol{x} - \boldsymbol{y}\|, \forall \boldsymbol{x}, \boldsymbol{y} \in \mathcal{X}. \tag{1}$$

The smallest possible $K$ in Equation (1) is the Lipschitz constant of $f$, denoted as $\mathrm{Lip}(f)$:

$$\mathrm{Lip}(f) = \sup_{\boldsymbol{x}, \boldsymbol{y} \in \mathcal{X}, \boldsymbol{x} \neq \boldsymbol{y}} \frac{\|f(\boldsymbol{x}) - f(\boldsymbol{y})\|}{\|\boldsymbol{x} - \boldsymbol{y}\|}. \tag{2}$$

In this context, $f$ is referred to as a $K$-Lipschitz function. *The Lipschitz bound essentially quantifies the maximum change in the output of a function corresponding to a unit-norm perturbation in its*

*input.* This makes the Lipschitz bound an important measure of a neural network's stability with respect to its input features. However, determining the exact Lipschitz bound can be computationally challenging. As discussed in Virmaux & Scaman (2018), finding the exact Lipschitz bound is shown to be NP-hard for deep models. Consequently, an upper bound is often sought as a practical alternative, and this upper bound is also referred to as the Lipschitz bound.

**Graph Neural Networks**   We assume a given graph $G = G(V, E)$, where $V$ denotes the set of nodes and $E$ denotes the set of edges. We will use $\boldsymbol{X} = \{\boldsymbol{x}_1, \boldsymbol{x}_2, \cdots, \boldsymbol{x}_N\} \subset \mathbb{R}^F$ to denote the $N$ node features in $\mathbb{R}^F$, as the input of any layer of a GNN. By abuse of notation, when there is no confusion, we also follow GNN literature and consider $\boldsymbol{X}$ as the $\mathbb{R}^{N \times F}$ matrix whose $i$-th row is given by $\boldsymbol{x}_i^\top$, $i = 1, \cdots, N$, though it unnecessarily imposes an ordering of the graph nodes.

GNNs are functions that operate on the adjacency matrix $\boldsymbol{A} \in \mathbb{R}^{N \times N}$ of a graph $G$. Specifically, an $L$-layer GNN can be defined as $f : \mathbb{R}^{N \times F^{\text{in}}} \to \mathbb{R}^{N \times F^{\text{out}}}$ that depends on $\boldsymbol{A}$. Formally, we have:

**Definition 1.** *An $L$-layer GNN is a function $f$ that can be expressed as a composition of $L$ message-passing layers $h^l$ and $L - 1$ activation functions $\rho^l$, as follows:*

$$f = h^L \circ \rho^{L-1} \circ \cdots \circ \rho^1 \circ h^1, \tag{3}$$

*where $h^l : \mathbb{R}^{F^{l-1}} \to \mathbb{R}^{F^l}$ is the $l$-th message-passing layer, $\rho^l : \mathbb{R}^{F^l} \to \mathbb{R}^{F^l}$ is the non-linear activation function in the $l$-th layer, and $F^{l-1}$ and $F^l$ denote the input and output feature dimensions for the $l$-th message-passing layer $h^l$, respectively. In addition, we set $l = 1, \cdots, L$.*

**Rank-based Individual Fairness of GNNs**   Rank-based Individual Fairness on GNNs (Dong et al., 2021) focuses on the relative order of instances rather than their absolute predictions. It ensures that similar instances, as measured by the similarity measure $S(\cdot, \cdot)$, receive consistent rankings or predictions. The criterion requires that if instance $i$ is more similar to instance $j$ than to instance $k$, then the predicted ranking of $j$ should be higher than that of $k$, *consistently*. This can be expressed as:

$$\text{if } S(\boldsymbol{x}_i, \boldsymbol{x}_j) > S(\boldsymbol{x}_i, \boldsymbol{x}_k), \text{ then } \boldsymbol{Y}_{ij} > \boldsymbol{Y}_{ik}, \tag{4}$$

where $\boldsymbol{Y}_{ij}$ and $\boldsymbol{Y}_{ik}$ denote the predicted rankings or predictions for instances $\boldsymbol{x}_j$ and $\boldsymbol{x}_k$, respectively, based on the input instance $\boldsymbol{x}_i$. The criterion ensures that the predicted rankings or predictions align with the relative similarities between instances, promoting fairness and preventing discriminatory predictions based on irrelevant factors.

## 3   ESTIMATING LIPSCHITZ BOUNDS ON GNNS FOR FAIRNESS

We estimate the Lipschitz bounds for GNNs, which is crucial for analyzing output perturbations induced by input biases: Initially, in § 3.1, we establish Lipschitz bounds for GNNs and provide closed-form formulas for these bounds. Next, in § 3.2, we derive the model's Jacobian matrix to facilitate Lipschitz bounds' efficient calculation for practical training feasibility. Lastly, in § 3.3, we use these bounds to explore individual fairness, demonstrating how model stability can ensure output consistency, aligning with rank-based individual fairness definition.

### 3.1   STABILITY OF THE MODEL OUTPUT

To analyze the stability of the model's output, we examine the Lipschitz bound of the Jacobian matrix of the GNN model. In this regard, we introduce the following lemma:

**Lemma 1.** *Given a function $g : \mathbb{R}^m \to \mathbb{R}^n$ with components $g_i$ for $i \in [n]$, the Lipschitz bound of $g$, denoted by $\mathrm{Lip}(g)$, is bounded above by the norm of the vector of the Lipschitz bounds of its components, that is:*

$$\mathrm{Lip}(g) \leq \|[\mathrm{Lip}(g_i)]_{i=1}^n\| \tag{5}$$

*where $\|\cdot\|$ represents the norm of a vector.*

Lemma 1 provides an inequality that relates the norm of the difference between two vector-valued functions, $g(x)$ and $g(y)$, to the norm of a vector composed of the component-wise differences of the functions evaluated at $x$ and $y$. Based on Lemma 1, we can now present the following theorem:

---

**Theorem 1.** *Let $\boldsymbol{Y}$ be the output of an L-layer GNN (denoted as $f(\cdot)$) with $\boldsymbol{X}$ as the input. Assuming the activation function (represented in $\rho(\cdot)$) is ReLU with a Lipschitz bound of $\mathrm{Lip}(\rho) = 1$, then the cumulative Lipschitz bound of the entire GNN, $\mathrm{Lip}(f)$, satisfies:*

$$\mathrm{Lip}(f) \leqslant \max_j \prod_{l=1}^{L} F^{l'} \left\| \left[ \mathcal{J}(h^l) \right]_j \right\|_\infty, \tag{6}$$

*where $F^{l'}$ represents the output dimension of the l-th message-passing layer, $j$ is the index of the node (e.g., j-th), and the vector $\left[ \mathcal{J}(h^l) \right] = \left[ \left\| \boldsymbol{J}_1(h^l) \right\|, \left\| \boldsymbol{J}_2(h^l) \right\|, \cdots, \left\| \boldsymbol{J}_{F^{l'}}(h^l) \right\| \right]$. Notably, $\boldsymbol{J}_i(h^l)$ denotes the i-th row of the Jacobian matrix of the l-th layer's input and output, and $\left[ \mathcal{J}(h^l) \right]_j$ is the vector corresponding to the j-th node in the l-th layer $h^l(\cdot)$.*

---

*Proof Sketch:*[1] We initiate with investigating the Lipschitz property of GNNs, considering 1-Lipschitz activation functions (e.g., the widely-used ReLU (Nair & Hinton, 2010). The hidden states of node feature $x_1$ and $x_2$ are represented as $z_1$ and $z_2$ respectively. The Lipschitz bound between these hidden states is calculated as $\frac{\|z_1 - z_2\|}{\|x_1 - x_2\|} = \frac{\|(h(x_1) - h(x_2))\|}{\|x_1 - x_2\|}$. Using the triangle inequality, this equation is further bounded by $\frac{\|z_1 - z_2\|}{\|x_1 - x_2\|} \leqslant \left\| \left[ \frac{h(x_1)_i - h(x_2)_i}{\|x_1 - x_2\|} \right]_{i=1}^{F'} \right\|$. The Lipschitz bound of individual elements is also analyzed, again using the triangle inequality. The proof then proceeds to analyze the Lipschitz bound of the individual elements using the operation of the l-th layer in $f(\cdot)$. Finally, the Lipschitz bound for the GNN is established as $\mathrm{Lip}(f) = \max_j \prod_{l=1}^{L} F^{l'} \left\| \left[ \mathcal{J}(h^l) \right]_j \right\|_\infty$. It establishes that the difference in the output $\boldsymbol{Y}$ is controlled by the Lipschitz bound of the GNN, $\mathrm{Lip}(f)$, and the difference in the input $\boldsymbol{X}$. The above analysis assesses the model output stability with respect to irrelevant biases learned layer-by-layer from the input, during the forward.

The aforementioned Theorem 1 provides the cumulative Lipschitz bound of the entire GNN based on the layer outputs and the corresponding Jacobian matrices. It establishes that the Lipschitz bound of a GNN regulates the magnitude of changes in the output induced by input biases, consequently guaranteeing the model output stability and fairness against irrelevant factors.

## 3.2 Simplifying Lipschitz Calculation of the GNN via Jacobian Matrix

The approach presented in Theorem 1 for estimating the Lipschitz bounds $\mathrm{Lip}(f)$ across different layers in the GNN $f(\cdot)$ requires unique explicit expressions for each component. This makes the process somewhat challenging. Nevertheless, this difficulty can be mitigated by computing the corresponding Jacobian matrices, as shown in Equation (6). By leveraging the values of each component in the Jacobian matrix, we can approximate the Lipschitz bounds in a straightforward manner. Additionally, the expression $\prod_{l=1}^{L} F^{l'} \left\| \left[ \mathcal{J}(h^l) \right]_j \right\|_\infty$ from Equation (6) provides valuable insights into the factors that influence the Lipschitz bounds, such as the dimensions of the output layer and the depth of the GNN layers.

However, considering the potential for multiple hierarchical layers in the network, their cumulative effect could lead to a significant deviation from the original bounds. Therefore, it is beneficial to consider the entire network as a single model and directly derive the Lipschitz bound from the input and output. In this subsection, we provide a *simplified* method to derive the Lipschitz bound in Equation (6) for facilitating fairness training. To achieve this, we then introduce the Jacobian matrix. Let $[\boldsymbol{J}_i]$ denote the Jacobian matrix of the i-th node, which can be calculated as:

$$[\boldsymbol{J}_i]_{F^{\mathrm{out}} \times F^{\mathrm{in}}} = \begin{bmatrix} \frac{\partial \boldsymbol{Y}_{i1}}{\partial \boldsymbol{X}_{i1}} & \frac{\partial \boldsymbol{Y}_{i1}}{\partial \boldsymbol{X}_{i2}} & \cdots & \frac{\partial \boldsymbol{Y}_{i1}}{\partial \boldsymbol{X}_{iF^{\mathrm{in}}}} \\ \frac{\partial \boldsymbol{Y}_{i2}}{\partial \boldsymbol{X}_{i1}} & \frac{\partial \boldsymbol{Y}_{i2}}{\partial \boldsymbol{X}_{i2}} & \cdots & \frac{\partial \boldsymbol{Y}_{i2}}{\partial \boldsymbol{X}_{iF^{\mathrm{in}}}} \\ \vdots & \vdots & \vdots & \vdots \\ \frac{\partial \boldsymbol{Y}_{iF^{\mathrm{out}}}}{\partial \boldsymbol{X}_{i1}} & \frac{\partial \boldsymbol{Y}_{iF^{\mathrm{out}}}}{\partial \boldsymbol{X}_{i2}} & \cdots & \frac{\partial \boldsymbol{Y}_{iF^{\mathrm{out}}}}{\partial \boldsymbol{X}_{iF^{\mathrm{in}}}} \end{bmatrix}_{F^{\mathrm{out}} \times F^{\mathrm{in}}}. \tag{7}$$

---

[1] Please refer to Appendix B for the complete proof due to space constraints.

We can define $[\boldsymbol{J}_i]_{F^{\text{out}} \times F^{\text{in}}} = \begin{bmatrix} \boldsymbol{J}_{i1}^\top, \boldsymbol{J}_{i2}^\top, \cdots, \boldsymbol{J}_{iF^{\text{out}}}^\top \end{bmatrix}^\top$, where $\boldsymbol{J}_{ij} = \begin{bmatrix} \frac{\partial \boldsymbol{Y}_{ij}}{\partial \boldsymbol{X}_{i1}}, \frac{\partial \boldsymbol{Y}_{ij}}{\partial \boldsymbol{X}_{i2}}, \cdots, \frac{\partial \boldsymbol{Y}_{ij}}{\partial \boldsymbol{X}_{iF^{\text{in}}}} \end{bmatrix}^\top$, capturing the local interactions of node $i$, rather than relational dynamics within the entire graph.[2] Then we let $\mathcal{J}_i = \begin{bmatrix} \|\boldsymbol{J}_{i1}\|, \|\boldsymbol{J}_{i2}\|, \cdots, \|\boldsymbol{J}_{iF^{\text{out}}}\| \end{bmatrix}^\top$, $\mathcal{J} = \begin{bmatrix} \mathcal{J}_1^\top, \mathcal{J}_2^\top, \cdots, \mathcal{J}_N^\top \end{bmatrix}^\top$. To analyze the Lipschitz bounds of the Jacobian matrix of output features for $N$ nodes, we define $\mathrm{LB}(\mathcal{J})$:

$$\mathrm{LB}(\mathcal{J}) = \begin{bmatrix} \mathcal{J}_1^\top \\ \mathcal{J}_2^\top \\ \vdots \\ \mathcal{J}_N^\top \end{bmatrix} = \begin{bmatrix} \mathcal{J}_{11} & \mathcal{J}_{12} & \cdots & \mathcal{J}_{1F^{\text{out}}} \\ J_{21} & \mathcal{J}_{22} & \cdots & \mathcal{J}_{2F^{\text{out}}} \\ \vdots & & & \\ \mathcal{J}_{N1} & \mathcal{J}_{N2} & \cdots & \mathcal{J}_{NF^{\text{out}}} \end{bmatrix}_{N \times F^{\text{out}}}, \tag{8}$$

where $\mathcal{J}_{ij} = \|\boldsymbol{J}_{ij}\|$. Based on the definition of $\mathrm{LB}(\mathcal{J})$, we can further establish the Lipschitz bound of the entire GNN model during training in the next subsection. To measure the scale of $\mathrm{LB}(\mathcal{J})$ of GNN $f(\cdot)$, we define a $\mathrm{Lip}(f)$ that satisfies

$$\mathrm{Lip}(f) = \|\mathrm{LB}(\mathcal{J})\|_{\infty,2}. \tag{9}$$

This calculation involves taking the $l_2\text{-}norm$ for each row of $\|\mathrm{LB}(\mathcal{J})\|_{\infty,2}$ and then taking the infinite norm for the entire $\|\mathrm{LB}(\mathcal{J})\|_{\infty,2}$. Now we have proposed an easy solution as Equation (9) to approximate $\prod_{l=1}^{L} F^{l'} \left\| \left[ \mathcal{J}(h^l) \right]_j \right\|_\infty$ for facilitating its feasible computation in practical training.

## 3.3 Illuminating GNN Fairness: A Rank-Based Perspective

Regularizing GNNs for input-output rank consistency, particularly in the context of *rank-based individual fairness*, is made possible by leveraging the Lipschitz bounds of GNNs. Specifically, we denote this Lipschitz bound as $\mathrm{Lip}(f) = \max_j \prod_{l=1}^{L} F^{l'} \left\| \left[ \mathcal{J}(h^l) \right]_j \right\|_\infty$. We aim to mitigate inherent biases in the training data and to advance individual fairness. We accomplish this by enforcing Lipschitz constraints on the model's output, thereby guarding against biases that are incrementally learned from the input during the forward pass. This ensures *consistency* between the ranking lists based on the similarity matrix of each node in the input graph $\boldsymbol{S_G}$ and the similarity matrix of the predicted outcome space $\boldsymbol{S_Y}$, as outlined in § 2.

**Algorithm 1** JacoLip: Simplified PyTorch-style Pseudocode for Lipschitz Bounds in Fairness-Oriented GNN Training

```
# model: graph neural network model
# Train model for N epochs
for X, A, target in dataloader:
    pred = model(X, A)
    ce_loss = CrossEntropyLoss(pred, target)

    # Compute Lipschitz bound for input
    jacobian = Jaco(X, pred)
    model_lip = Lip(jacobian) # Eq.(8)
    cum_lip = norm(model_lip) # Eq.(9)

    # Optimize model with Lipschitz bound
    loss = ce_loss + u * cum_lip
    loss.backward()
    optimizer.step()
```

We propose a plug-and-play solution, termed *JacoLip*, that integrates effortlessly with existing fairness-focused GNN training pipelines. Algorithm 1 provides a detailed PyTorch-style pseudocode for this approach. The method involves training the GNN model for a predefined number of epochs, while computing the Lipschitz bound for the model output using gradients and norms of the input features. This Lipschitz bound is then incorporated as a regularization term in the loss function to keep the model's output within constrained boundaries. Our approach efficiently mitigates individual biases and enhances the fairness of GNNs without incurring significant computational overhead. This "nearly-free" fairness regularization is facilitated by PyTorch's built-in gradient functions, thereby achieving the benefits of Lipschitz regularization without substantial additional computational burden.

## 4 Experiments

Here, we perform two major experiments to validate the Lipschitz bounds discussed in the previous section: *(1)* We use Lipschitz bounds to constrain the output consistency of GNNs, aiming to enhance rank-based individual fairness in node classification and link prediction tasks; *(2)* We examine the effects of Lipschitz bounds on the gradients and weights of GNNs, particularly concerning biases induced through training dynamics. Additional experiments can be found in Appendix C.

---

[2]Please refer to Appendix B.4 for more discussions.

### 4.1 SETUP

**Datasets**   We conduct experiments on six real-world datasets commonly used in prior work on rank-based individual fairness (Dong et al., 2021). These include one citation network (`ACM` (Tang et al., 2008)) and two co-authorship networks (`Co-author-CS` and `Co-author-Phy` (Shchur et al., 2018)) for node classification, and three social networks (`BlogCatalog` (Tang & Liu, 2009), `Flickr` (Huang et al., 2017), and `Facebook` (Leskovec & Mcauley, 2012)) for link prediction. We adhere to the public train/val/test splits from Dong et al. (2021). Dataset statistics and details are in Appendix E.

**Backbones**   We use two widely-adopted GNN architectures for each downstream learning task: Graph Convolutional Network (GCN) (Kipf & Welling, 2017) and Simplifying Graph Convolutional Network (SGC) (Wu et al., 2019) for node classification, and GCN and Variational Graph Auto-Encoders (GAE) (Kipf & Welling, 2016) for link prediction. Model details are in Appendix D.

**Baselines**   In contrast to prior research on group fairness in graph embeddings (e.g., (Bose & Hamilton, 2019; Rahman et al., 2019)), which often focuses on fairness for subgroups defined by specific protected attributes, our work centers on individual fairness (Dong et al., 2021) without reliance on such attributes. Therefore, methods for group fairness are beyond the scope of our comparison. To assess the efficacy of our approach in achieving individual fairness, we benchmark it against three significant baselines tailored specifically for rank-based individual fairness:

- *Redress* (Dong et al., 2021): This method proposes a rank-based framework to enhance the individual fairness of GNNs, integrating both utility maximization and fairness promotion into a joint end-to-end training framework.
- *InFoRM* (Kang et al., 2020): Originally designed for conventional graph mining tasks such as PageRank and Spectral Clustering, InFoRM is based on the Lipschitz condition. We adapt it to GNNs by combining its individual fairness loss with the unity loss of the GNN backbone for end-to-end optimization.
- *PFR* (Lahoti et al., 2019): PFR focuses on learning fair representations for individual fairness and outperforms traditional approaches in this regard. As PFR is more of a pre-processing strategy and not tailored for graph data, we apply it to the input node features.

**Evaluation Metrics**   We employ two key metrics for evaluating rank-based individual fairness: classification accuracy (*Acc.*) for node classification tasks, and the area under the receiver operating characteristic curve (*AUC*) for link prediction tasks. For assessing individual fairness, we use the widely adopted ranking metric *NDCG@k* (Järvelin & Kekäläinen, 2002), which measures the similarity between the rankings generated from $S_Y$ (result similarity matrix) and $S_G$ (oracle similarity matrix). Average *NDCG@k* values are reported across all nodes, with $k = 10$.

**Implementation Details**   The learning rate is set at 0.01 for all tasks. For models based on GCN and SGC, we use two layers with 16 hidden units each. For GAE-based models, we employ three graph convolutional layers, with the first two layers having 32 and 16 hidden units, respectively. Adam is used as the optimizer (Kingma & Ba, 2015). Further details, including code, dataset splits, and hyperparameter settings, are available in Appendix D. Our code has been released at `https://github.com/chunhuizng/lipschitz-fairness`.

### 4.2 EFFECT OF LIPSCHITZ BOUNDS ON RANK-BASED INDIVIDUAL FAIRNESS IN GRAPHS

The experiments validate the efficacy of Lipschitz bounds for enhancing individual fairness in GNNs, specifically from a ranking perspective. Results are consolidated in Tables 1 and 2 for the node classification and link prediction tasks, respectively.

In Table 1, our method, JacoLip, yields promising results, outperforming the baselines in achieving a superior trade-off between accuracy and fairness: *(i)* When applied to Vanilla models such as GCN or SGC, JacoLip enhances fairness performance (measured by NDCG@10) while maintaining comparable accuracy. This result substantiates that the plug-and-play nature of Lipschitz bound regularization effectively mitigates bias during the training of standard GNN backbones, thereby fostering a balanced trade-off between accuracy and fairness; *(ii)* Additionally, when integrated with the existing fairness-centric algorithm Redress, JacoLip further constrains irrelevant biased factors during training and marginally improves the trade-off between accuracy and fairness across all datasets and backbones.

Table 1: Evaluation on node classification task: comparing under accuracy and NDCG. Higher performance in both metrics indicates a better trade-off. Results are in percentages, and averaged values and standard deviations are computed from five runs. The improvement is within brackets. We bold the best result and underline the runner-up.

| Data | Model | Fair Alg. | Feature Similarity | | Structural Similarity | |
|---|---|---|---|---|---|---|
| | | | utility: Acc.↑ | fairness: NDCG@10↑ | utility: Acc.↑ | fairness: NDCG@10↑ |
| ACM | GCN | Vanilla (Kipf & Welling, 2017) | **72.49±0.6** | 47.33±1.0 | **72.49±0.6** | 25.42±0.6 |
| | | InFoRM (Kang et al., 2020) | 68.03±0.3(−6.15%) | 39.79±0.3(−15.9%) | 69.13±0.5(−4.64%) | 12.02±0.4(−52.7%) |
| | | PFR (Lahoti et al., 2019) | 67.88±1.1(−6.36%) | 31.20±0.2(−34.1%) | 69.00±0.7(−4.81%) | 23.85±1.3(−6.18%) |
| | | Redress (Dong et al., 2021) | 71.75±0.4(−1.02%) | 49.13±0.4(+3.80%) | 72.03±0.9(−0.63%) | 29.09±0.4(+14.4%) |
| | | **JacoLip** (on Vanilla) | 72.37±0.3(−0.16%) | 49.80±0.3(+5.26%) | 71.97±0.3(−0.71%) | 27.91±0.7(+9.79%) |
| | | **JacoLip** (on Redress) | 71.92±0.2(−0.78%) | **53.62±0.6(+13.3%)** | 72.05±0.5(−0.60%) | **31.80±0.4(+25.1%)** |
| | SGC | Vanilla (Wu et al., 2019) | 68.40±1.0 | 55.75±1.1 | 68.40±1.0 | 37.18±0.6 |
| | | InFoRM (Kang et al., 2020) | 68.81±0.5(+0.60%) | 48.25±0.5(−13.5%) | 66.71±0.6(−2.47%) | 28.33±0.6(−23.8%) |
| | | PFR (Lahoti et al., 2019) | 67.97±0.7(−0.62%) | 34.71±0.1(−37.7%) | 67.78±0.1(−0.91%) | 37.15±0.6(−0.08%) |
| | | Redress (Dong et al., 2021) | 67.16±0.2(−1.81%) | 58.64±0.4(+5.18%) | 67.77±0.4(−0.92%) | 38.95±0.1(+4.76%) |
| | | **JacoLip** (on Vanilla) | **73.84±0.2(+7.95%)** | 62.00±0.2(+11.21%) | 69.28±0.3(+1.29%) | 38.36±0.4(+3.17%) |
| | | **JacoLip** (on Redress) | 72.36±0.4(+5.79%) | **69.22±0.5(+24.16%)** | 72.52±0.5(+6.02%) | **41.07±0.3(+10.5%)** |
| CS | GCN | Vanilla (Kipf & Welling, 2017) | 90.59±0.3 | 50.84±1.2 | **90.59±0.3** | 18.29±0.8 |
| | | InFoRM (Kang et al., 2020) | 88.66±1.1(−2.13%) | 53.38±1.6(+5.00%) | 87.55±0.9(−3.36%) | 19.18±0.9(+4.87%) |
| | | PFR (Lahoti et al., 2019) | 87.51±0.7(−3.40%) | 37.12±0.9(−27.0%) | 86.16±0.2(−4.89%) | 11.98±1.3(−34.5%) |
| | | Redress (Dong et al., 2021) | **90.70±0.2(+0.12%)** | 55.01±1.9(+8.20%) | 89.16±0.3(−1.58%) | 21.28±0.3(+16.4%) |
| | | **JacoLip** (on Vanilla) | 90.68±0.3(+0.90%) | 55.35±0.2(+8.87%) | 89.23±0.5(−1.50%) | 21.82±0.2(+19.3%) |
| | | **JacoLip** (on Redress) | 90.63±0.3(+0.40%) | **68.20±0.4(+34.2%)** | 89.21±0.1(−1.52%) | **31.82±0.4(+74.1%)** |
| | SGC | Vanilla (Wu et al., 2019) | 87.48±0.8 | 74.00±0.1 | 87.48±0.8 | 32.36±0.3 |
| | | InFoRM (Kang et al., 2020) | 88.07±0.1(+0.67%) | 74.29±0.1(+0.39%) | 88.65±0.4(+1.34%) | 32.37±0.4(+0.03%) |
| | | PFR (Lahoti et al., 2019) | 88.31±0.1(+0.94%) | 48.40±0.1(−34.6%) | 84.34±0.3(−3.59%) | 28.87±0.9(−10.8%) |
| | | Redress (Dong et al., 2021) | 90.01±0.2(+2.89%) | 76.60±0.1(+3.51%) | 89.35±0.1(+2.14%) | 34.24±0.2(+5.81%) |
| | | **JacoLip** (on Vanilla) | **90.23±0.2(+3.14%)** | 74.63±0.2(+0.85%) | 89.53±0.6(+2.34%) | 32.83±0.3(+1.45%) |
| | | **JacoLip** (on Redress) | 90.12±0.3(+3.02%) | **77.01±0.1(+4.07%)** | **89.80±0.2(+2.65%)** | **34.89±0.5(+7.82%)** |
| Phy | GCN | Vanilla (Kipf & Welling, 2017) | **94.81±0.2** | 34.83±1.1 | **94.81±0.2** | 1.57±0.1 |
| | | InFoRM (Kang et al., 2020) | 89.33±0.8(−5.78%) | 31.25±0.0(−10.3%) | 94.46±0.2(−0.37%) | 1.77±0.0(+12.7%) |
| | | PFR (Lahoti et al., 2019) | 89.74±0.5(−5.35%) | 24.16±0.4(−30.6%) | 87.26±0.2(−7.96%) | 1.20±0.1(−23.6%) |
| | | Redress (Dong et al., 2021) | 94.63±0.7(−0.19%) | 43.64±0.5(+25.3%) | 93.94±0.3(−0.92%) | 1.93±0.1(+22.9%) |
| | | **JacoLip** (on Vanilla) | 94.60±0.2(−0.22%) | 37.33±0.5(+7.18%) | 93.99±0.4(−0.86%) | 1.87±0.2(+19.1%) |
| | | **JacoLip** (on Redress) | 94.50±0.2(−0.32%) | **49.37±0.3(+41.75%)** | 93.86±0.9(−1.00%) | **2.72±0.1(+73.3%)** |
| | SGC | Vanilla (Wu et al., 2019) | **94.45±0.2** | 49.63±0.1 | **94.45±0.2** | 3.61±0.1 |
| | | InFoRM (Kang et al., 2020) | 92.01±0.1(−2.58%) | 43.87±0.2(−11.6%) | 94.27±0.3(−0.19%) | 3.64±0.0(+0.83%) |
| | | PFR (Lahoti et al., 2019) | 89.74±0.3(−4.99%) | 28.54±0.1(−42.5%) | 89.73±0.3(−5.00%) | 2.62±0.1(−27.4%) |
| | | Redress (Dong et al., 2021) | 94.30±0.1(−0.16%) | 53.40±0.1(+7.60%) | 93.94±0.2(−0.54%) | 4.03±0.0(+11.6%) |
| | | **JacoLip** (on Vanilla) | 94.20±0.3(−0.26%) | 50.70±0.4(+2.16%) | 93.58±0.2(−0.92%) | 3.80±0.6(+5.26%) |
| | | **JacoLip** (on Redress) | 93.28±0.1(−1.24%) | **59.20±0.6(+19.3%)** | 93.99±1.1(−0.49%) | **4.30±0.5(+19.1%)** |

Similar trends are observed for the link prediction task, as seen in Table 2. Here, the performance of Vanilla models (GCN or GAE), InFoRM, PFR, Redress, and JacoLip is evaluated based on AUC for utility and NDCG@10 for fairness. Across both tasks, JacoLip consistently shows either competitive or superior performance when compared to the baselines. These improvements are achieved with minimal computational overhead, as JacoLip efficiently calculates Lipschitz bounds using PyTorch's built-in gradient functionality, further detailed in § 3.3 and Table 5.

## 4.3 IMPACT OF LIPSCHITZ BOUNDS ON TRAINING DYNAMICS

We further analyze the impact of Lipschitz bounds through the optimization process and explore its dynamical interactions with weight parameters, gradient, fairness, and accuracy in Figure 1:

*For the nonlinear GCN model*, our proposed method, JacoLip, imposes beneficial regularizations on gradients. Specifically, during the initial epochs, JacoLip effectively stabilizes gradient magnitudes, resulting in higher accuracy (e.g., approximately 20% improvement in feature similarity and around 40% in structural similarity) compared to the Redress baseline. Concurrently, JacoLip maintains superior fairness metrics (NDCG), exceeding the baseline Redress; *For the linear SGC model*, JacoLip similarly achieves a favorable trade-off between fairness and accuracy relative to the Redress baseline. Initially, JacoLip on SGC shows better fairness (e.g., approximately 0.2 NDCG on feature similarity) than the Redress baseline, with only a minor decrease in accuracy. As training progresses, the accuracy of JacoLip on SGC tends to match or even surpass that of Redress.

In summary, the observed trade-off between accuracy and fairness during the training phase can be attributed to the constraining effect of Lipschitz bounds on gradient optimization. The higher expressivity of the nonlinear GCN model renders it more susceptible to loss of consistency in input-output similarity ranking, crucial for fairness. Conversely, the simpler linear model more readily preserves this consistency, and here, JacoLip prioritizes accuracy. These insights elucidate the

Table 2: Evaluation on link prediction tasks: comparing under AUC and NDCG.

| Data | Model | Fair Alg. | Feature Similarity | | Structural Similarity | |
|---|---|---|---|---|---|---|
| | | | utility: AUC↑ | fairness: NDCG@10↑ | utility: AUC↑ | fairness: NDCG@10↑ |
| Blog | GCN | Vanilla (Kipf & Welling, 2017) | 85.87±0.1 | 16.73±0.1 | 85.87±0.1 | 32.47±0.5 |
| | | InFoRM (Kang et al., 2020) | 79.85±0.6(−7.01%) | 15.57±0.2(−6.93%) | 84.00±0.1(−2.18%) | 26.18±0.3(−19.4%) |
| | | PFR (Lahoti et al., 2019) | 84.25±0.2(−1.89%) | 16.37±0.0(−2.15%) | 83.88±0.0(−2.32%) | 29.60±0.4(−8.84%) |
| | | Redress (Dong et al., 2021) | 86.49±0.8(+0.72%) | 17.66±0.2(+5.56%) | 86.25±0.3(+0.44%) | 34.62±0.7(+6.62%) |
| | | **JacoLip (on Vanilla)** | **86.51±0.2(+0.74%)** | 17.70±0.6(+5.79%) | **86.90±0.5(+1.67%)** | 35.00±0.4(+7.79%) |
| | | **JacoLip (on Redress)** | 85.91±0.2(+0.04%) | **18.02±0.6(+7.71%)** | 86.84±0.5(+1.13%) | **35.85±0.4(+10.4%)** |
| | GAE | Vanilla (Kipf & Welling, 2016) | 85.72±0.1 | 17.13±0.1 | 85.72±0.1 | 41.99±0.4 |
| | | InFoRM (Kang et al., 2020) | 80.01±0.2(−6.66%) | 16.12±0.2(−5.90%) | 82.86±0.0(−3.34%) | 27.29±0.3(−35.0%) |
| | | PFR (Lahoti et al., 2019) | 83.83±0.1(−2.20%) | 16.64±0.0(−2.86%) | 83.87±0.1(−2.16%) | 35.91±0.4(−14.5%) |
| | | Redress (Dong et al., 2021) | 84.67±0.9(−1.22%) | 18.19±0.1(+6.19%) | **86.36±1.5(+0.75%)** | 43.51±0.7(+3.62%) |
| | | **JacoLip (on Vanilla)** | **85.75±0.4(+0.03%)** | 17.96±0.5(+4.85%) | 85.86±0.5(+0.16%) | 42.20±0.3(+0.50%) |
| | | **JacoLip (on Redress)** | 85.70±0.4(−0.02%) | **18.34±0.5(+7.06%)** | 86.31±0.5(+0.69%) | **43.60±0.3(+3.83%)** |
| Flickr | GCN | Vanilla (Kipf & Welling, 2016) | 92.20±0.3 | 13.10±0.2 | 92.20±0.3 | 22.35±0.3 |
| | | InFoRM (Kang et al., 2020) | 91.39±0.0(−0.88%) | 11.95±0.1(−8.78%) | 91.73±0.1(−0.51%) | 23.28±0.6(+4.16%) |
| | | PFR (Lahoti et al., 2019) | 91.91±0.1(−0.31%) | 12.94±0.0(−1.22%) | 91.86±0.2(−0.37%) | 19.80±0.4(−11.4%) |
| | | Redress (Dong et al., 2021) | 91.38±0.1(−0.89%) | 13.58±0.3(+3.66%) | 92.67±0.2(+0.51%) | 28.45±0.5(+27.3%) |
| | | **JacoLip (on Vanilla)** | **92.75±0.3(+0.59%)** | 13.74±0.4(+4.89%) | 92.54±0.1(+0.37%) | 26.61±0.4(+19.1%) |
| | | **JacoLip (on Redress)** | 92.53±0.3(+0.35%) | **14.37±0.4(+9.69%)** | 92.69±0.1(+0.53%) | **28.65±0.4(+28.2%)** |
| | GAE | Vanilla (Kipf & Welling, 2016) | 89.98±0.1 | 12.77±0.0 | 89.98±0.1 | 23.58±0.2 |
| | | InFoRM (Kang et al., 2020) | 88.76±0.7(−1.36%) | 12.07±0.1(−5.48%) | **91.51±0.2(+1.70%)** | 15.78±0.3(−33.1%) |
| | | PFR (Lahoti et al., 2019) | **90.30±0.1(+0.36%)** | 12.12±0.1(−5.09%) | 90.10±0.1(+1.33%) | 20.46±0.3(−13.2%) |
| | | Redress (Dong et al., 2021) | 89.45±0.5(−0.59%) | 14.24±0.1(+11.5%) | 89.52±0.3(−0.51%) | 29.83±0.2(+26.5%) |
| | | **JacoLip (on Vanilla)** | 89.88±0.3(−0.11%) | 14.37±0.1(+12.53%) | 89.95±0.2(−0.03%) | 28.74±0.5(+21.9%) |
| | | **JacoLip (on Redress)** | 89.92±0.3(−0.06%) | **14.85±0.1(+16.29%)** | 89.56±0.2(−0.46%) | **30.04±0.5(+28.7%)** |
| Facebook | GCN | Vanilla (Kipf & Welling, 2017) | 95.60±1.7 | 23.07±0.2 | 95.60±1.7 | 16.55±1.1 |
| | | InFoRM (Kang et al., 2020) | 90.26±0.1(−5.59%) | 23.23±0.3(+0.69%) | **96.66±0.6(+1.11%)** | 15.18±0.7(−8.28%) |
| | | PFR (Lahoti et al., 2019) | 87.11±1.2(−8.88%) | 21.83±0.2(−5.37%) | 94.87±1.9(−0.76%) | 19.53±0.5(+18.0%) |
| | | Redress (Dong et al., 2021) | **96.49±1.6(+0.93%)** | 29.60±0.1(+28.3%) | 92.66±0.4(−3.08%) | 27.73±1.1(+67.5%) |
| | | **JacoLip (on Vanilla)** | 96.21±0.2(+0.63%) | 29.47±0.3(+27.7%) | 95.46±0.9(−0.14%) | 26.60±0.1(+60.7%) |
| | | **JacoLip (on Redress)** | 96.11±0.2(+0.53%) | **30.07±0.3(+30.3%)** | 92.76±0.9(−2.97%) | **28.64±0.1(+73.1%)** |
| | GAE | Vanilla (Kipf & Welling, 2016) | **98.54±0.0** | 26.75±0.1 | **98.54±0.0** | 27.03±0.1 |
| | | InFoRM (Kang et al., 2020) | 90.50±0.4(−8.16%) | 22.77±0.2(−14.9%) | 95.03±0.1(−3.56%) | 15.38±0.2(−43.1%) |
| | | PFR (Lahoti et al., 2019) | 96.91±0.1(−1.65%) | 23.52±0.1(−12.1%) | 98.28±0.0(−0.26%) | 22.89±0.3(−15.3%) |
| | | Redress (Dong et al., 2021) | 95.98±1.5(−2.60%) | 28.43±0.3(+6.28%) | 94.07±1.7(−4.54%) | **33.53±0.2(+24.0%)** |
| | | **JacoLip (on Vanilla)** | 97.40±0.1(−1.16%) | 27.44±0.6(+2.58%) | 97.02±1.1(−1.54%) | 30.90±0.5(+14.3%) |
| | | **JacoLip (on Redress)** | 96.10±0.1(−2.48%) | **28.46±0.6(+6.39%)** | 94.22±1.1(−4.38%) | 31.62±0.5(+17.1%) |

dynamic behavior of model training under Lipschitz bound constraints, illustrating how JacoLip can enhance fairness while retaining competitive accuracy.

## 5 RELATED WORKS

**Lipschitz Bounds in Deep Models.** Prior research on Lipschitz bounds has primarily focused on specific types of neural networks incorporating convolutional or attention layers (Zou et al., 2019; Terris et al., 2020; Kim et al., 2021; Araujo et al., 2021). In the context of GNNs, (Dasoulas et al., 2021) introduced a Lipschitz normalization method for self-attention layers in GATs. More recently, (Gama & Sojoudi, 2022) estimated the filter Lipschitz bound using the infinite norm of a matrix. In contrast, the Lipschitz matrix in our study follows a distinct definition and employs different choices of norm types. Additionally, our objective is to enhance the stability of GNNs against unfair biases, which is not clearly addressed in the aforementioned works. We refer to Appendix A for vital details.

**Fair Graph Learning.** Fair graph learning is a relatively open field (Wu et al., 2021; Dai & Wang, 2021; Buyl & De Bie, 2020). Some existing approaches address fairness concerns through fairness-aware augmentations or adversarial training. For instance, Fairwalk (Rahman et al., 2019) is a random walk-based algorithm that aims to mitigate fairness issues in graph node embeddings. Adversarial training is employed in approaches like Compositional Fairness (Bose & Hamilton, 2019) to disentangle learned embeddings from sensitive features. Information Regularization (Liao et al., 2021) utilizes adversarial training to minimize the marginal difference between vertex representations. In addition, (Palowitch & Perozzi, 2020) improves group fairness by ensuring that node embeddings lie on a hyperplane orthogonal to sensitive features. However, there remains ample room for further exploration in rank-based individual fairness (Dong et al., 2021), which is the focus of our work.

**Understanding Learning on Graphs.** Various approaches have emerged to understand the underlying patterns in the graph data and its components. Explanatory models for relational/graph learning (Ying et al., 2019; Huang et al., 2022; Yuan et al., 2021; Chen et al., 2023) provide insights into the relationship between a model's predictions and elements in graphs. These works shed light on how

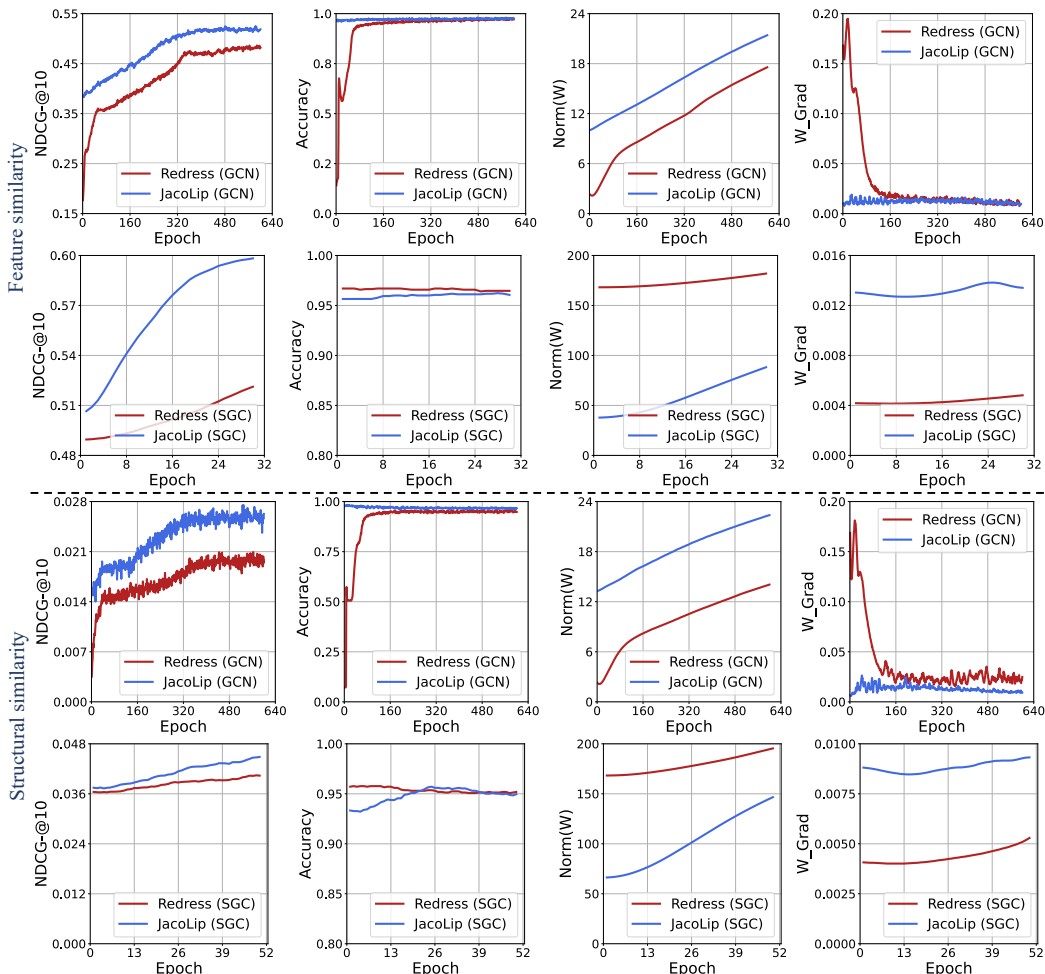

Figure 1: Study of the Lipschitz bounds' impact on model training for rank-based individual fairness. We perform experiments on the co-author-Physics dataset using both nonlinear (GCN) and linear (SGC) models. The training dynamics are assessed by monitoring the NDCG, accuracy, weight norm, and weight gradient as the number of epochs increases. *Upper two rows*: Metrics under feature similarity; *Lower two rows*: Metrics under structural similarity.

local elements or node characteristics influence the decision-making process of GNNs. However, our work differs in that we investigate the impact of Lipschitz bounds on practical training dynamics, rather than focusing on trained/fixed-parameter model inference or the influence of local features on GNN decision-making processes.

## 6 CONCLUSIONS

We have investigated the use of Lipschitz bounds for promoting individual fairness in GNNs from a ranking perspective. We conducted a thorough analysis of the theoretical properties of Lipschitz bounds and their relation to rank-based individual fairness. Building on this analysis, we introduced JacoLip, a fairness solution that incorporates Lipschitz bound regularization into the GNN training process. To assess the efficacy of JacoLip, we conducted extensive experiments on real-world datasets for both node classification and link prediction tasks. The results consistently demonstrate that JacoLip effectively constrains bias during training, thereby enhancing fairness performance while preserving accuracy.

ETHICS STATEMENT

Our research on Lipschitz bounds for GNNs carries societal implications. On the positive side, our work enhances the stability and interpretability of GNNs, contributing to safer AI systems across various domains. Improved fairness and interpretability can help mitigate biases and ensure more equitable decision-making outcomes. On the other hand, the advancement of GNNs also brings forth concerns about potential misuse and unintended consequences. Like any deep learning technology, GNNs have the potential to unfairly impact certain groups or perpetuate existing biases.

We underscore the importance of fair training procedures, stringent privacy safeguards, and responsible deployment and monitoring of GNNs. In this manner, our work serves as a foundational contribution to GNN research, emphasizing the need to consider broader impacts and potential harms while implementing suitable mitigation strategies.

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

## A    DETAILED RELATED WORK

**In the field of Lipschitz Bounds in Neural Networks**, the stability of neural networks in relation to Lipschitz bounds was initially spotlighted by Szegedy et al. (2014), which postulated that larger Lipschitz bounds could cause network instability. An insightful proposition was the use of the product of the spectral norms across layers to upper bound the Lipschitz bound. Following this foundation, a slew of research (Virmaux & Scaman, 2018; Fazlyab et al., 2019; Jordan & Dimakis, 2020; Latorre et al., 2020; Huang et al., 2021b) fine-tuned the estimation of Lipschitz bounds, ushering in optimized frameworks for tighter bounds. Delving into particular network architectures, studies have explored the Lipschitz properties of convolutional layers and attention mechanisms (Zou et al., 2019; Terris et al., 2020; Araujo et al., 2021; Kim et al., 2021).

**For Lipschitz Bounds in GNNs**, the exploration of Lipschitz bounds took specific turns. Dasoulas et al. (2021) pioneered a normalization technique rooted in Lipschitz bounds for self-attention layers in Graph Attention Networks (GAT). This innovative method proved instrumental in training deeper GAT models by curbing gradient explosion issues. On another front, Gama & Sojoudi (2021) paved a new path by estimating the Lipschitz bound of filters through the infinite norm of matrices, with each matrix element epitomizing the Lipschitz bound for a respective position filter.

**About Lipschitz, Fairness, and GNNs**, A significant stride in the context of Lipschitz bounds and fairness was the introduction of a linear bound propagation method (Shi et al., 2022). This methodology adeptly estimates Lipschitz bounds across different network sections. However, it's imperative to underscore its primary design catering to Euclidean datasets, making its direct application to non-Euclidean graph data and specific fairness issues a challenge. Notably, a linkage between fairness and stability in GNNs was etched by Agarwal et al. (2021), which presented an innovative framework enabling GNNs to learn fair and stable graph representations. However, it primarily focuses on attribute group fairness, emphasizing fairness for explicitly labeled sensitive attributes. This fundamentally differs from our focus on individual fairness and no requirements on explicitly labeled sensitive attributes (e.g., gender, race), making direct comparisons infeasible. Dwork et al. (2012) conducted an in-depth investigation into the realm of fairness within classification, presenting a comprehensive framework that encompassed a task-specific metric as well as an algorithm used to optimize utility while retaining fairness constraints. However, it centered on conventional statistical learning models, and did not extend to deep models or graph models.

**Our Distinction in Context**: (i) while the aforementioned studies about Lipschitz, we address the notable gap in integrating Lipschitz bounds within GNN training, particularly *focusing on individual fairness on graph structures*; (ii) furthermore, our emphasis on *facilitating the efficient calculation of Lipschitz bounds for practical training feasibility* by deriving the Jacobian matrix of the model for practical training and the interpretability aspects, sets our method apart.

### A.1    COMPARISON BETWEEN INDIVIDUAL FAIRNESS AND GROUP FAIRNESS

The concepts of individual and group fairness are fundamental in the domain of machine learning ethics, particularly when designing and evaluating algorithms for fair decision-making. Both concepts aim to address fairness concerns, but they do so from different perspectives and with distinct implications. While group fairness relies on annotated sensitive attributes, individual fairness does not. The distinction between individual and group fairness in terms of reliance on annotated attributes highlights a fundamental difference in how these fairness paradigms conceptualize and address fairness.

**Individual Fairness**    This concept is centered around the notion of treating similar individuals similarly. In a machine learning context, it implies that if two individuals are similar with respect to the attributes relevant to a decision-making process (e.g., loan approval, job recruitment), they should be treated in a comparable manner by the algorithm. Operationalizing individual fairness often involves defining a suitable metric of similarity between individuals, which can be challenging. This metric should capture all the relevant aspects that justify similar treatment. One of the main challenges is the subjective nature of defining similarity. What constitutes "similar" in one context or for one set of stakeholders might not be agreed upon universally. There's also a computational challenge in ensuring this kind of fairness at scale, as it may require pairwise comparisons among individuals.

Individual fairness has (1) No Need for Annotated Attributes: Individual fairness typically does not rely on explicitly annotated attributes, especially sensitive attributes like race, gender, or age. Instead, it focuses on the idea of treating similar individuals similarly, where similarity is often defined in the context of the specific task or decision process. Instead, it relies on (2) Implicit Attributes: The concept of similarity in individual fairness is based on implicit attributes derived from the context or the nature of the task. These attributes are usually not explicitly labeled but are inferred from the data or the specific decision-making scenario.

**Group Fairness** This approach focuses on ensuring fairness across predefined groups, typically defined by explicit sensitive attributes like race, gender, or age. Group fairness is concerned with statistical measures and often aims for equal treatment or outcomes across these groups. Common criteria include demographic parity, equal opportunity, and equalized odds. Group fairness is easier to quantify and implement than individual fairness as it relies on statistical measures (e.g., ensuring that selection rates for a job are equal across different gender groups). Group fairness is often applied in large-scale decision-making scenarios where societal or policy-level fairness concerns are paramount, such as in credit scoring or hiring processes. A significant challenge with group fairness is the risk of oversimplification. By focusing on broad groups, it might overlook nuances and individual-level disparities within these groups. Additionally, it can sometimes lead to unfair outcomes for individuals when trying to balance statistics at the group level.

Group fairness (1) Relies on Annotated Attributes: Group fairness explicitly relies on annotated attributes, often focusing on sensitive or protected attributes. These are explicit labels in the dataset, such as race, gender, or other demographic information. It uses (2) Explicit Categories: In group fairness, individuals are categorized based on these explicit attributes, and fairness is measured by evaluating the outcomes or treatments across these predefined groups. This approach simplifies the fairness problem by reducing it to a series of statistical measures (like demographic parity, equal opportunity, etc.) across these groups. While this simplification aids in quantification and implementation, it may overlook individual-level disparities and nuanced differences within groups.

## A.2 GNN and Fairness – A Simplified, Easy-to-understand Overview

**Graph Neural Networks (GNNs)** GNNs are designed to process graph or relational data. Unlike conventional data structures, graphs consist of nodes and edges, which represent entities and their relationships. GNNs excel in capturing complex patterns in such data by aggregating information from neighboring nodes, making them useful for tasks like classification, prediction, and clustering. It is popularly used on diverse real-world applications, such as social media modeling (Wu et al., 2022; Qian et al., 2022; Tian et al., 2023b; Guo et al., 2023; Li et al., 2023; Wen et al., 2022a), from which the platform also derives societal concerns such as public safety, fairness, and privacy (Wen et al., 2022b; Liu et al., 2023; Ouyang et al., 2023; Liu et al., 2024).

**Ranking-based Individual Fairness** The concept of individual fairness in the context of ranking is centered around the principle that individuals who are similar, based on defined attributes, should be accorded comparable rankings. This principle is particularly pertinent in the domain of GNNs, where the objective often involves ranking nodes — such as users, items, or entities — based on a blend of their intrinsic features and the nature of their interconnections within the graph.

In this approach, the focus is on the relative positioning of individuals or nodes within the ranking order. The key is not the absolute scores or ratings assigned to each individual, but rather ensuring that the rank order is fair and equitable. This means that if two nodes are deemed similar in terms of their attributes or their roles within the network, this similarity should be reflected in how they are ranked. Whether it's in generating search results, recommendations, or categorizing items, the ranking-based fairness approach strives to preserve the integrity of this relative ordering, maintaining consistency across various representations or subsets of the data. It's about upholding a fair and justifiable hierarchy that resonates with the underlying similarity and relationship patterns among the nodes in the graph.

**Lipschitz Condition and Individual Fairness in our work** The Lipschitz condition is a mathematical concept that, in this context, can be used to enforce a form of individual fairness. A function is said to be Lipschitz continuous if there exists a bound $L$ such that for any two points $x$ and $y$, the

difference in the function's outputs is at most $L$ times the distance between $x$ and $y$. In simpler terms, similar inputs lead to similar outputs with a bound on how different the outputs can be.

When applied to ranking, a Lipschitz condition can ensure that if two nodes are similar (close in the graph structure or feature space), their differences in ranking (output of the GNN) are limited. This prevents wildly different rankings for similar nodes, which contributes to individual fairness.

## B    PROOFS

### B.1    NOTATIONS

We use the following notations throughout the paper. Sets are denoted by $\{\}$ and vectors by $()$. For $n \in \mathbb{N}$, we denote $[n] = \{1, \cdots, n\}$. Scalars are denoted by regular letters, lowercase bold letters denote vectors, and uppercase bold letters denote matrices. For instance, $\boldsymbol{x} = (x_1, \cdots, x_n)^\top \in \mathbb{R}^n$ and $\boldsymbol{X} = [X_{ik}]_{i \in [n], k \in [m]} \in \mathbb{R}^{n \times m}$. For any vector $\boldsymbol{x} \in \mathbb{R}^n$, we use $\|\boldsymbol{x}\|$ to denote its $\ell_2$-norm: $\|\boldsymbol{x}\| = \left(\sum_{i=1}^n x_i^2\right)^{1/2}$. For any matrix $\boldsymbol{X} \in \mathbb{R}^{n \times m}$, we use $\boldsymbol{X}_{i,:}$ to denote its $i$-th row and $\boldsymbol{X}_{:,k}$ to denote its $k$-th column. The $(\infty, 2)$-norm of $\boldsymbol{X}$ is denoted by $\|\boldsymbol{X}\|_{\infty,2} = \max_{i \in [n]} \|\boldsymbol{X}_{i,:}\|$. Given a graph $G = G(V, E)$ with ordered nodes, we denote its adjacency matrix by $\boldsymbol{A}$ such that $A_{ij} = 1$ if $\{i, j\} \in E$ and $A_{ij} = 0$ otherwise. When it is clear from the context, we use $\boldsymbol{X} \in \mathbb{R}^{N \times F}$ to denote a feature matrix whose $i$-th row corresponds to the features of the $i$-th node, and the $j$-th column represents the features across all nodes for the $j$-th attribute. We denote the output of the GNN as $\boldsymbol{Y} \in \mathbb{R}^{N \times C}$, where $N$ is the number of nodes and $C$ is the number of output classes.

### B.2    PROOF OF LEMMA 1 IN § 3.1

> **Lemma 1.** *Given a function $g : \mathbb{R}^m \to \mathbb{R}^n$ with components $g_i$ for $i \in [n]$, the Lipschitz bound of $g$, denoted by $\mathrm{Lip}(g)$, is bounded above by the norm of the vector of the Lipschitz bounds of its components, that is:*
> $$\mathrm{Lip}(g) \leq \|[\mathrm{Lip}(g_i)]_{i=1}^n\| \tag{10}$$
> *where $\|\cdot\|$ represents the norm of a vector, and $[\mathrm{Lip}(g_i)]_{i=1}^n$ represents a vector whose components are the Lipschitz bounds of the functions $g_i$.*

*Proof.* We begin by observing that

$$\frac{\|g(\boldsymbol{x}) - g(\boldsymbol{y})\|}{\|\boldsymbol{x} - \boldsymbol{y}\|} = \frac{\|[g_i(\boldsymbol{x}) - g_i(\boldsymbol{y})]_{i=1}^n\|}{\|\boldsymbol{x} - \boldsymbol{y}\|}$$
$$= \left\| \left[ \frac{|g_i(\boldsymbol{x}) - g_i(\boldsymbol{y})|}{\|\boldsymbol{x} - \boldsymbol{y}\|} \right]_{i=1}^n \right\|. \tag{11}$$

Furthermore, for each $i \in [n]$, $\dfrac{|g_i(\boldsymbol{x}) - g_i(\boldsymbol{y})|}{\|\boldsymbol{x} - \boldsymbol{y}\|} \leq \mathrm{Lip}(g_i)$. Therefore, we can write

$$\mathrm{Lip}(g) = \sup_{\boldsymbol{x} \neq \boldsymbol{y}} \frac{\|g(\boldsymbol{x}) - g(\boldsymbol{y})\|}{\|\boldsymbol{x} - \boldsymbol{y}\|}$$
$$= \sup_{\boldsymbol{x} \neq \boldsymbol{y}} \left\| \left[ \frac{|g_i(\boldsymbol{x}) - g_i(\boldsymbol{y})|}{\|\boldsymbol{x} - \boldsymbol{y}\|} \right]_{i=1}^n \right\| \tag{12}$$
$$\leq \sup_{\boldsymbol{x} \neq \boldsymbol{y}} \|[\mathrm{Lip}(g_i)]_{i=1}^n\| = \|[\mathrm{Lip}(g_i)]_{i=1}^n\|,$$

this completes the proof. $\qquad\square$

In the above proof of Lemma 1, we start by rewriting the norm of the difference between $g(\boldsymbol{x})$ and $g(\boldsymbol{y})$ divided by the norm of $\boldsymbol{x} - \boldsymbol{y}$ as a norm of a vector containing the component-wise differences of $g_i(\boldsymbol{x})$ and $g_i(\boldsymbol{y})$ divided by the norm of $\boldsymbol{x} - \boldsymbol{y}$ for each $i$. We then observe that for each $i$, the absolute value of $g_i(\boldsymbol{x}) - g_i(\boldsymbol{y})$ divided by the norm of $\boldsymbol{x} - \boldsymbol{y}$ is bounded by the Lipschitz bound $\mathrm{Lip}(g_i)$. Hence, the Lipschitz bound of $g$ is bounded by the norm of the vector $[\mathrm{Lip}(g_i)]_{i=1}^n$. This establishes the inequality in Lemma 1.

### B.3 PROOF OF THEOREM 1 IN § 3.1

**Theorem 1.** *Let $\mathbf{Y}$ be the output of an L-layer GNN (represented in $f(\cdot)$) with $\mathbf{X}$ as the input. Assuming the activation function (represented in $\rho(\cdot)$) is ReLU with a Lipschitz bound of $\mathrm{Lip}(\rho) = 1$, then the cumulative Lipschitz bound of the entire GNN, denoted as $\mathrm{Lip}(f)$, satisfies the following inequality:*

$$\mathrm{Lip}(f) \leqslant \max_j \prod_{l=1}^{L} F^{l'} \left\| \left[ \mathcal{J}(h^l) \right]_j \right\|_\infty , \tag{13}$$

*where $F^{l'}$ represents the output dimension of the l-th message-passing layer, $j$ is the index of the node (e.g., $j$-th), and the vector $\left[ \mathcal{J}(h^l) \right] = \left[ \left\| \mathbf{J}_1(h^l) \right\|, \left\| \mathbf{J}_2(h^l) \right\|, \cdots, \left\| \mathbf{J}_{F^{l'}}(h^l) \right\| \right]$. Notably, $\mathbf{J}_i(h^l)$ denotes the i-th row of the Jacobian matrix of the l-th layer's input and output, and $\left[ \mathcal{J}(h^l) \right]_j$ is the vector corresponding to the j-th node in the l-th layer $h^l(\cdot)$.*

The inequality in Theorem 1 constrains the cumulative Lipschitz bound of the entire GNN based on the layer outputs and Jacobian matrices. It is derived as follows:

*Proof.* We begin by examining the Lipschitz property of the GNN. Let $\mathbf{Y}$ denote the output of an $L$-layer GNN with input $\mathbf{X}$. Assuming the commonly used ReLU activation function as the non-linear layer $\rho(\cdot)$, we have $\mathrm{Lip}(\rho) = 1$. First, we consider the Lipschitz bound between the hidden states of two nodes output by any message-passing layer $h(\cdot)$ in $f(\cdot)$. Let $z_1$ and $z_2$ represent the hidden states of node features $x_1$ and $x_2$, respectively. The Lipschitz bound between these hidden states is given by:

$$\frac{\|z_1 - z_2\|}{\|x_1 - x_2\|} = \frac{\| (h(x_1) - h(x_2)) \|}{\|x_1 - x_2\|}. \tag{14}$$

By applying the triangle inequality, we obtain:

$$\frac{\|z_1 - z_2\|}{\|x_1 - x_2\|} \leqslant \left\| \left[ \frac{h(x_1)_i - h(x_2)_i}{\|x_1 - x_2\|} \right]_{i=1}^{F'} \right\|, \tag{15}$$

Next, we consider the Lipschitz bound between individual elements of the hidden states. Let $f(x_1)$ and $f(x_2)$ represent the hidden state matrices for inputs $x_1$ and $x_2$, respectively. By again applying the triangle inequality, we have:

$$\frac{\|z_1 - z_2\|}{\|x_1 - x_2\|} \leqslant \left\| F' \times \max_i \frac{h(x_1)_i - h(x_2)_i}{\|x_1 - x_2\|} \right\|, \tag{16}$$

let's focus on the Lipschitz bound of the individual elements, $\frac{h(x_1)_i - h(x_2)_i}{\|x_1 - x_2\|}$: here, $f(x)_i$ denotes the $i$-th column of the matrix $f(x)$. We denote $h^l(\cdot)$ as the operation of the $l$-th message-passing layer in $f(\cdot)$, then by applying the triangle inequality and leveraging the Lipschitz property of the ReLU activation function, we have:

$$\frac{\|f(x_1)_{j,1} - f(x_2)_{j,2}\|}{\|x_{j,1} - x_{j,2}\|} \leqslant \prod_{l=1}^{L} F^{l'} \left\| \left[ \mathcal{J}(h^l) \right]_j \right\|_\infty , \tag{17}$$

where $x_{j,1}$ and $x_{j,1}$ denote features of $j$-th node's in $x_1$ and $x_2$, respectively, and $\left[ \mathcal{J}(h^l) \right]_j$ represents the $j$-th node's the Jacobian matrix of the $l$-th message-passing layer. Therefore, the Lipschitz bound for the GNN can be expressed as:

$$\mathrm{Lip}(f) = \max_j \prod_{l=1}^{L} F^{l'} \left\| \left[ \mathcal{J}(h^l) \right]_j \right\|_\infty . \tag{18}$$

In summary, we have shown that for any two input samples $x_1$ and $x_2$, the Lipschitz bound of the GNN, denoted as $\mathrm{Lip}(f)$, satisfies:

$$\|\mathbf{Y}_1 - \mathbf{Y}_2\| \leqslant \mathrm{Lip}(f) \|\mathbf{X}_1 - \mathbf{X}_2\| , \tag{19}$$

where $\boldsymbol{Y}$ denotes the output of the GNN for inputs $\boldsymbol{X}$. This inequality implies that the Lipschitz bound $\mathrm{Lip}(f)$ controls the magnitude of changes in the output based on input biases/perturbations. Therefore, we have established the following result:

$$\|\boldsymbol{Y}_1 - \boldsymbol{Y}_2\| \leqslant \prod_{l=1}^{L} F^{l'} \left\| \left[ \mathcal{J}(h^l) \right]_j \right\|_\infty \|\boldsymbol{X}_1 - \boldsymbol{X}_2\|. \tag{20}$$

This inequality demonstrates that the Lipschitz bound of the GNN, $\mathrm{Lip}(f)$, controls the magnitude of the difference in the output $\boldsymbol{Y}$ based on the difference in the input $\boldsymbol{X}$. It allows us to analyze the stability of the model's output with respect to input perturbations. $\qquad\square$

### B.4 THE LOCALITY IN LIPSCHITZ COMPUTATION

For local Lipschitz bound computation on graph data, we clarify that the Jacobian matrix is a 2D tensor, and we don't consider the case of Equation (7) where $i \neq j$:

- If $\frac{\partial Y_{j1}}{\partial X_{i1}}$ where $i \neq j$, then the computed Lipschitz bound will not be a **local** Lipschitz bound (as in our paper with $\frac{\partial Y_{i1}}{\partial X_{i1}}$) but rather a **global** Lipschitz bound, which considers the Lipschitz relationships for all nodes. However, as discussed in §3 of Dong et al. (2021): solving for the global Lipschitz bounds of a 2-layer MLP is an NP-Hard problem. Therefore, even considering this facet for simple models (like GCN and SGC) would lead to a significant increase in running time, rendering the computation of Lipschitz bounds impractical.

- However, we contend that **global** Lipschitz bounds do not align with graph individual fairness (we enable similar individuals in the input space to maintain their similarity in the output space, ensuring individual fairness. It is achieved by that: we constrain the expansion of data in the input space to the output space, thereby minimizing the impact of variations in input data on the output space, which is shaped as a Lipschitz constraint problem): First, if $i \neq j$, the equations are considering the Lipschitz bound for the entire relational network. In this case, the Lipschitz bound takes into account each pair of nodes within the network (i.e., each single one node must consider its interactions with all other nodes). This doesn't consider minor perturbations; rather, the perturbation can be understood as the difference between different nodes. For example, the dimension of each node takes into account the impact of all other nodes (again, in this scenario, the complexity is extremely high because the dimension of each node needs to account for all dimensions of all other nodes, leading to an astonishing amount of computation). Under these circumstances, it is the global Lipschitz bound, which doesn't consider the locality of our GCN's local message passing (unlike our local Lipschitz bound) on graph data; Second, in Dong et al. (2021), the ranking-based individual fairness computes similarity based on the **nearest k neighbors**, which is a **local** similarity measure and more aligned with our local Lipschitz.

- In this context, when $i = j$, the **local** Lipschitz bounds are more aligned with the objective, as there is no need to maintain pairwise similarity between all nodes in the output space: a node is solely influenced by its neighboring nodes. When nodes are similar, meaning they have a high degree of similarity, the difference in features between two nodes can be approximated by a small perturbation. We only need to consider how this perturbation affects the output, which can be quantified using gradients. If the gradient is steep, even a small perturbation can lead to a significant impact.

Moreover, the Jacobian matrix serves as a suitable measure for the Lipschitz bounds on each node and dimension, and frameworks like PyTorch provide mechanisms to expedite its computation. This approach reduces algorithmic complexity, making it more adaptable, while also aligning with the principles of individual fairness.

### B.5 HOW LIPSCHITZ BOUNDS CAN GUARANTEE RANKING-BASED INDIVIDUAL FAIRNESS?

In the context of individual fairness on graphs, the relationship between ranking-based individual fairness and Lipschitz continuity is established through Lipschitz constraints on the GNN model. The Lipschitz continuity ensures that the model's predictions are not overly sensitive to small changes in node embeddings. Let's define the two concepts and their relationship mathematically:

**Lipschitz Property**    In the GNN model $f : V \rightarrow \mathbb{R}^d$, Lipschitz continuity constrains the change in model output concerning small **changes/biases** in input (node embeddings). A function $f$ is Lipschitz continuous if there exists a bound $K$ such that:

$$\|f(z_i) - f(z_j)\| \leq K\|z_i - z_j\|, \tag{21}$$

for any two input points $z_i$ and $z_j$. Here, $\|\cdot\|$ denotes a norm (e.g., $L2$ norm), and $K$ is the Lipschitz bound.

**Ranking-based Individual Fairness**    Based on prior work (Dong et al., 2021; Kang et al., 2020), this concept in GNNs focuses on maintaining the rank-ordering of pairwise node **similarities** pre and post-training. This is achieved by a ranking loss function $L(\cdot)$ that penalizes discrepancies between predicted pairwise distances and original similarities as:

$$\text{Ranking Loss: } \mathcal{L}_{\text{rank}} = \sum_{(v_i, v_j) \in \mathcal{P}} L(S(v_i, v_j), D(z_i, z_j)), \tag{22}$$

where $\mathcal{P}$ is the set of all node pairs, and $L(\cdot)$ measures the discrepancy between predicted pairwise distance $D(z_i, z_j)$ and original similarity $S(v_i, v_j)$.

**The Exact Relationship**    By imposing a Lipschitz constraint on the GNN, the model's output becomes less sensitive to minor variations (i.e., **minor biases**) in node embeddings, thereby reinforcing ranking-based individual fairness. The mathematical connection can be expressed as:

$$\text{RankLoss}(S(v_i, v_j), D(z_i, z_j)) \propto L(f(z_i), f(z_j)), \tag{23}$$

where $\text{RankLoss}(S(v_i, v_j), D(z_i, z_j))$ is the ranking loss function, measuring the discrepancy between predicted pairwise distances $D(z_i, z_j)$ and original similarities $S(v_i, v_j)$ in the output space.

Enforcing a Lipschitz constraint on the GNN's model parameters or architecture inherently promotes the preservation of the rank-ordering of node similarities during training. This approach drives the GNN to uphold individual fairness by ensuring that nodes with similar attributes receive comparable rankings in the output space, resulting in fair and unbiased predictions for each node.

In summary, the principle of Lipschitz continuity in GNNs is directly related to the concept of ranking-based individual fairness. It achieves this by moderating the model's response to small variations in the embeddings of nodes.

## C   ADDITIONAL EXPERIMENTS

### C.1   NODE CLASSIFICATION TASK: EVALUATED UNDER ACCURACY AND ERROR

According to Table 3, on node classification tasks, JacoLip consistently shows a competitive or improved trade-off between accuracy and error compared to the baselines, highlighting the effectiveness of the Lipschitz bound in promoting individual fairness on graphs.

### C.2   CONTRASTING WITH LIPSCHITZ METHODS TAILORED FOR OTHER DOMAINS

Although there are dozens of Lipschitz methods of fundamental statistical models (Dwork et al., 2012) or deep models Shi et al. (2022); Agarwal et al. (2021), it is essential to acknowledge their distinctiveness, making comparisons less straightforward.

#### C.2.1   CONSIDERATION ON LIPSCHITZ METHODS FOR CNN/MLP

When contrasting our approach with previous Lipschitz regularization studies for general CNN/MLP, we use Shi et al. (2022) as an example, and we highlight the following points:

First, the approach in Shi et al. (2022) does not directly use existing Jacobian matrices. It introduces a concept called Clarke Jacobian, leading to explicit expressions of the bounds. This method requires utilizing the neural network's forward pass, necessitating computations of input and output at each step of the network, making the process complex. It primarily focuses on computing the Lipschitz

Table 3: Evaluation on node classification tasks: comparing under accuracy and error.

| Data | Model | Fair Alg. | Feature Similarity | | Structural Similarity | |
|---|---|---|---|---|---|---|
| | | | utility: Acc.↑ | fairness: Err.@10↑ | utility: Acc.↑ | fairness: Err.@10↑ |
| ACM | GCN | Vanilla (Kipf & Welling, 2017) | 72.49±0.6 | 75.70±0.6 | **72.49±0.6** | 37.55±0.4 |
| | | InFoRM (Kang et al., 2020) | 67.65±1.0(−6.68%) | 73.49±0.5(−2.92%) | 65.91±0.2(−9.07%) | 19.96±0.6(−46.8%) |
| | | PFR (Lahoti et al., 2019) | 68.48±0.6(−5.53%) | 76.28±0.1(0.77%) | 70.22±0.7(−3.13%) | 36.54±0.4(−2.69%) |
| | | Redress (Dong et al., 2021) | **73.46±0.2(+1.34%)** | 82.27±0.1(+8.68%) | 71.87±0.4(−0.86%) | 43.74±0.0(+16.5%) |
| | | **JacoLip** (on Vanilla) | 72.80±0.2(+4.27%) | **82.88±0.1(+9.48%)** | 72.30±0.4(−0.26%) | 39.28±0.2(+4.61%) |
| | | **JacoLip** (on Redress) | 71.05±0.4(+2.00%) | 82.21±0.3(+8.60%) | 71.92±0.3(−0.79%) | **46.13±0.3(+22.85%)** |
| | SGC | Vanilla (Wu et al., 2019) | 68.40±1.0 | 80.06±0.1 | 68.40±1.0 | 45.95±0.3 |
| | | InFoRM (Kang et al., 2020) | 67.96±0.5(−0.64%) | 75.63±0.5(−5.53%) | 66.16±0.6(−3.27%) | 39.79±0.1(−13.4%) |
| | | PFR (Lahoti et al., 2019) | 67.69±0.4(−1.04%) | 76.80±0.1(−4.07%) | 66.69±0.3(−2.50%) | 46.99±0.5(+2.26%) |
| | | Redress (Dong et al., 2021) | 66.51±0.3(−2.76%) | 82.32±0.3(+2.82%) | 67.10±0.7(−1.90%) | 49.02±0.2(+4.76%) |
| | | **JacoLip** (on Vanilla) | **74.04±0.2(+8.25%)** | 82.73±0.7(+5.18%) | **72.91±0.9(+3.33%)** | 48.64±0.2(+5.85%) |
| | | **JacoLip** (on Redress) | 69.91±0.1(+2.21%) | **85.22±0.4(+6.45%)** | 71.27±0.3(+4.20%) | **52.01±0.4(+13.23%)** |
| CS | GCN | Vanilla (Kipf & Welling, 2017) | **90.59±0.3** | 80.41±0.1 | **90.59±0.3** | 26.69±1.3 |
| | | InFoRM (Kang et al., 2020) | 88.37±0.9(−2.45%) | 80.63±0.6(+0.27%) | 87.10±0.9(−3.85%) | 29.68±0.6(+11.2%) |
| | | PFR (Lahoti et al., 2019) | 87.62±0.2(−3.28%) | 76.26±0.1(−5.16%) | 85.66±0.7(−5.44%) | 19.80±1.4(−25.8%) |
| | | Redress (Dong et al., 2021) | 90.06±0.5(−0.59%) | 83.24±0.2(+3.52%) | 89.91±0.2(−0.86%) | 32.42±1.6(+21.5%) |
| | | **JacoLip** (on Vanilla) | 90.41±0.4(−0.20%) | 82.57±0.1(+2.69%) | 89.12±0.1(−1.62%) | 32.80±0.6(+22.74%) |
| | | **JacoLip** (on Redress) | 90.30±0.3(−0.32%) | **88.11±0.3(+9.58%)** | 89.93±0.2(−0.73%) | **42.50±0.4(+59.24%)** |
| | SGC | Vanilla (Wu et al., 2019) | 87.48±0.8 | 90.58±0.1 | 87.48±0.8 | 43.28±0.2 |
| | | InFoRM (Kang et al., 2020) | 87.31±0.5(−0.19%) | 90.64±0.1(+0.07%) | 88.21±0.4(+0.83%) | 44.37±0.1(+0.21%) |
| | | PFR (Lahoti et al., 2019) | 87.95±0.2(+0.54%) | 79.85±0.2(−11.8%) | 86.93±0.1(−0.63%) | 38.83±0.8(−10.3%) |
| | | Redress (Dong et al., 2021) | 90.48±0.2(+3.43%) | 92.03±0.1(+1.60%) | 90.39±0.1(+3.33%) | 45.81±0.0(+5.85%) |
| | | **JacoLip** (on Vanilla) | 90.71±0.3(+3.69%) | 90.75±0.4(+0.19%) | 90.34±1.0(+3.27%) | 43.92±0.3(+1.48%) |
| | | **JacoLip** (on Redress) | **92.22±0.2(+5.42%)** | **92.22±0.4(+1.81%)** | 90.54±0.3(+3.50%) | **46.39±0.5(+7.19%)** |
| Phy | GCN | Vanilla (Kipf & Welling, 2017) | **94.81±0.2** | 73.25±0.3 | **94.81±0.2** | 2.58±0.1 |
| | | InFoRM (Kang et al., 2020) | 88.67±0.7(−6.48%) | 73.80±0.6(+0.75%) | 94.68±0.2(−0.14%) | 2.45±0.1(−5.04%) |
| | | PFR (Lahoti et al., 2019) | 88.79±0.2(−6.35%) | 73.22±0.4(+0.10%) | 89.69±1.0(−5.40%) | 1.67±0.1(−35.3%) |
| | | Redress (Dong et al., 2021) | 93.71±0.1(−1.16%) | 80.23±0.1(+9.53%) | 93.91±0.4(−0.95%) | 3.22±0.3(+22.9%) |
| | | **JacoLip** (on Vanilla) | 93.71±0.2(−1.16%) | 78.64±1.1(+7.36%) | 94.75±0.3(−0.06%) | 2.75±0.6(+6.80%) |
| | | **JacoLip** (on Redress) | 93.79±0.8(−1.08%) | **82.60±0.3(+12.70%)** | 93.98±0.3(−0.88%) | **4.0±0.1(+55.43%)** |
| | SGC | Vanilla (Wu et al., 2019) | 94.45±0.2 | 77.48±0.2 | 94.45±0.2 | 4.50±0.1 |
| | | InFoRM (Kang et al., 2020) | 92.06±0.2(−2.53%) | 75.13±0.4(−3.03%) | 94.27±0.1(−0.19%) | 4.44±0.0(−1.33%) |
| | | PFR (Lahoti et al., 2019) | 87.39±1.2(−7.47%) | 73.42±0.2(−5.24%) | 89.16±0.3(−5.60%) | 3.41±0.2(−24.2%) |
| | | Redress (Dong et al., 2021) | **94.81±0.2(+0.38%)** | 79.57±0.2(+2.70%) | **94.54±0.1(+0.10%)** | 4.98±0.1(+10.7%) |
| | | **JacoLip** (on Vanilla) | 94.43±0.7(−0.02%) | 78.82±0.8(+1.73%) | 94.09±0.6(−0.38%) | 4.75±0.2(+5.56%) |
| | | **JacoLip** (on Redress) | 94.78±0.1(+0.35%) | **82.21±0.2(+6.10%)** | 93.00±1.3(−1.54%) | **5.45±0.1(+1.90%)** |

bounds of the network, striving to find a tight bound. However, the paper does not provide specific application results for fairness, leaving doubts about its effectiveness in fairness-related contexts.

Second, in contrast, our method is simpler and leverages PyTorch's built-in automatic differentiation mechanism for all computations, requiring no additional computational cost. Furthermore, our approach treats the entire network model as a whole, disregarding the internal computation flow of the network. It only requires the model's inputs and outputs, making it easy and convenient to use. Importantly, our Lipschitz method is designed to cater to the application context of individual fairness. Our goal is not solely to find a tight bound, but rather to integrate the Lipschitz bound as a regularization term into the training process. By continuously constraining the Lipschitz bound during training, we ensure individual fairness. The underlying objectives of the two methods are distinct: while the approach in Shi et al. (2022) aims for tight bounds, our Lipschitz bound serves the purpose of ensuring individual fairness. In the context of fairness, achieving a tight bound is not the primary concern. Instead, our focus is on preserving fairness in the predictions, where the exact tightness of the bound is less crucial to GNN fairness training.

Third, extending our investigation to GNNs using the code provided by Shi et al. (2022), we computed local Lipschitz bounds. Our experimental results are tabulated at Table 4, revealing the utility and fairness performances of both methods. Notably, the accuracy-fairness tradeoff in Shi et al. (2022) is lower than our approach. This observation echoes the earlier speculation of an application mismatch, as outlined in the preceding points. Moreover, we also infer that the graph structure's (adjacency matrix) influence on GNN's forward pass necessitates specialized treatment during Jacobian bound calculations. This realization prompts a more profound inquiry—one that should be taken into account when contemplating potential applications of Shi et al. (2022) to graph individual fairness.

Finally, beyond the performance benefits from our fairness design's alignment with graph structures (as Equation (8)'s consideration on graphs), our methodology also champions the efficient com-

Table 4: Comparisons on the FB dataset with the GAE backbone.

|  | Feat. Simi. | | Struct. Simi. | |
| --- | --- | --- | --- | --- |
|  | utility: AUC↑ | fairness: NDCG@10↑ | utility: AUC↑ | fairness: NDCG@10↑ |
| **Shi et al. (2022)** | 94.40 | 27.40 | 96.28 | 22.90 |
| **JacoLip (Ours)** | 97.40 | 27.44 | 97.02 | 30.90 |

putation of Lipschitz bounds for non-Euclidean models. While Dwork et al. (2012) pioneered the regularization of Lipschitz bounds for fairness, it could not cater to graph data as efficiently as our research, since it is proposed on traditional (relatively) simple statistical models. Similarly, although Shi et al. (2022) does compute a tight Lipschitz bound for CNN/MLP, our method emphasizes empirical individual-fairness predictions over theoretical tightness. This is evidenced by our prior experiments and this preliminary time-per-iteration comparison with Shi et al. (2022) at Table 5:

Table 5: Time-per-iteration comparisons on the FB dataset.

| Method | Practical Training Feasibility | | Time/Iter. |
| --- | --- | --- | --- |
|  | Graph Structure | Efficiency | |
| GCN without Lipschitz | N.A. | N.A. | 0.2841s |
| Ours | Yes | Yes | 0.2972s |
| Shi et al. (2022) | No | No | 3.2407s |

### C.2.2 CONSIDERATION ON LIPSCHITZ METHODS ON ATTRIBUTED GRAPHS

Furthermore, Agarwal et al. (2021) established a link between fairness and stability, introducing a novel framework for GNNs to acquire both fair and stable graph representations. However, it pursued group fairness to enhance **attribute fairness** for **explicitly labeled sensitive attributes**, resulting in a distinct definition from our emphasis on individual fairness within regular graphs with no explicitly labeled sensitive attributes. Consequently, our research and that of Agarwal et al. (2021) cannot be appropriately compared in the same experimental scenarios due to fundamental differences in motivation and application contexts.

### C.3 SENSITIVITY OF THE $\mu$ FOR THE LIPSCHITZ REGULARIZATION TERM

To provide a comprehensive analysis for understanding how varying hyperparameters impact the Lipschitz bound of the GNN, we experimented with different values of the hyperparameter $\mu$, which directly influences the Lipschitz bound of the model. A larger value of $\mu$ correlates with a smaller Lipschitz bound, making it a pivotal factor in our approach. Below are Table 6 and Table 7 illustrating the effects of varying $\mu$ on the GCN model applied to the Node Classification task on the Co-author-CS dataset. These tables present the outcomes in terms of AUC and NDCG metrics:

Table 6: Feature Similarity: GCN on Co-author-CS dataset with varying $\mu$

| $\mu$ | 0 | 0.1 | 0.01 | 0.001 | 0.0001 | 0.00001 | 0.000001 |
| --- | --- | --- | --- | --- | --- | --- | --- |
| NDCG@10↑ | 50.84 | 37.67 | 48.98 | 63.56 | 68.32 | 67.91 | 66.26 |
| AUC↑ | 90.59 | 60.18 | 89.96 | 90.96 | 90.20 | 90.23 | 90.18 |

The results from these tables clearly show that both AUC and NDCG metrics are sensitive to variations in the Lipschitz bound, which is controlled by $\mu$. Notably, when $\mu$ is relatively large, resulting in a smaller Lipschitz bound, there is a significant impact on both performance and fairness metrics. As $\mu$ decreases, allowing the Lipschitz bound of the model to approach that of the original model, we observe a convergence of both performance and fairness metrics towards those of the original model.

This analysis highlights the delicate balance that must be maintained between fairness and performance in our approach. The data reveals that the metric of fairness is more responsive to changes in $\mu$ compared to performance. Therefore, identifying an optimal Lipschitz bound that adequately meets both fairness and performance criteria becomes paramount in our methodology.

Table 7: Structure Similarity: GCN on Co-author-CS dataset with varying $\mu$

| $\mu$ | 0 | 0.1 | 0.01 | 0.001 | 0.0001 | 0.00001 | 0.000001 |
|---|---|---|---|---|---|---|---|
| NDCG@10↑ | 18.29 | 4.11 | 10.66 | 19.62 | 31.82 | 26.82 | 19.42 |
| AUC↑ | 90.34 | 33.46 | 86.98 | 89.25 | 89.12 | 89.18 | 89.51 |

## C.4 COMPUTATIONAL OVERHEAD

We provide a detailed computational comparison using the FB dataset. This dataset comprises 4,039 nodes, 88,234 edges, and 1,406 attributes, making it a representative choice for our analysis. Table 8 below presents a comparative analysis of time and GPU memory usage for various methods applied to the FB dataset. The results from this comparison demonstrate that our proposed method incurs no additional computational overhead when contrasted with the standard training procedures. These findings offer a more concrete empirical evidence to support it.

Table 8: Time and GPU memory usage of different methods on FB dataset.

| Method | Lipschitz Tightness | Graph Structure | Efficiency | Time/Iter. | GPU Memory-Usage |
|---|---|---|---|---|---|
| GCN (standard) | N.A. | N.A. | N.A. | 0.2841s | 414.8 MiB |
| Lipschitz originally for CNN/MLP Shi et al. (2022) | Restrictive | No | No | 3.2407s | 2130.1 MiB |
| Layered Lipschitz Computing for GNN Jia et al. (2023) | Restrictive | Yes | No | 3.8541s | 3280.7 MiB |
| Ours (GCN + JacoLip) | Restrictive | Yes | Yes | 0.2972s | 420.0 MiB |

## C.5 HOW JACOLIP AVOIDS MEMORY AND COMPUTE-INTENSIVE OPERATIONS?

A key to our approach is the mitigation of the memory and computational challenges commonly associated with computing the norms of the Jacobian matrix. We achieve this efficiency by avoiding the direct computation of the Jacobian matrix and consequently, the need for second-order derivative calculations.

Specifically, our method capitalizes on the capability to compute the norm of the Jacobian matrix without the necessity to explicitly form the Jacobian itself. This approach eliminates the requirement for gradient of gradients computation. The key segment of our code is in Listing 1:

```python
for i in range(out.shape[1]):
    # Directional derivative vectors are initialized as zero vectors
    v = torch.zeros_like(out)
    v[:, i] = 1

    # Compute gradients with respect to the inputs, not the model parameters
    gradients = autograd.grad(outputs=out, inputs=input, grad_outputs=v,
                              create_graph=True, retain_graph=True, only_inputs=True)[0]

    # We only compute the norm of these gradients, which is a first-order operation
    grad_norm = torch.norm(gradients, dim=1).unsqueeze(dim=1)
    lip_mat.append(grad_norm)
```

Listing 1: Python code for computing gradients and norms

This code snippet demonstrates the computation of the gradient concerning the inputs, which is a first-order derivative. While we set the `create_graph=True` parameter, which allows for higher-order derivative computations, our method only utilizes this for first-order derivative calculations. The rationale is that our Lipschitz bound approximation demands only the norms of these gradients, **not** the gradients of these norms. Consequently, the computational complexity of our method remains at the first-order derivative level, with only a marginal increase compared to standard training procedures.

Furthermore, we implement an efficient batching of gradient norm computations and do not retain the computational graph of these norms. This approach ensures that the memory footprint does not scale with the size of the Jacobian, but increases linearly with the output size. Given that the output size in relational datasets typically corresponds to a small dimension (i.e., the number of classes), the increase in memory requirement is minimal and manageable.

Additionally, our method leverages the inherent efficiencies of PyTorch's autograd system. This system dynamically allocates and deallocates memory during the training process, optimizing gradient computations and ensuring minimal memory usage by retaining only necessary gradients.

In summary, our approach computes the Lipschitz bound without incurring the significant computational overhead associated with second-order derivatives. Our empirical results, supported by the provided code, affirm that our method does not result in substantial increases in training time or GPU memory usage when compared with standard training procedures.

## D   MODEL CARD

The hyper-parameters for our method across all datasets are listed in Table 9. For fair comparisons, we follow the default settings of Redress (Dong et al., 2021).

Table 9: Hyperparameters used in our experiments.

| Hyperparameters | Node Classification | | | Link Prediction | | |
|---|---|---|---|---|---|---|
| | ACM | Coauthor-CS | Coauthor-Phy | Blog | Flickr | Facebook |
| Hyperparameters w.r.t. the GCN model | | | | | | |
| # Layers | 2 | 2 | 2 | 2 | 2 | 2 |
| Hidden Dimension | $[16, 9]$ | $[16, 15]$ | $[16, 5]$ | $[32, 16]$ | $[32, 16]$ | $[32, 16]$ |
| Activation | ReLU used for all datasets | | | | | |
| Dropout | 0.03 | 0.03 | 0.03 | 0.00 | 0.00 | 0.00 |
| Optimizer | AdamW with $1e - 5$ weight decay | | | | | |
| Pretrain Steps | 300 | 300 | 300 | 200 | 200 | 200 |
| Training Steps | 150 | 200 | 200 | 60 | 100 | 50 |
| Learning Rate | 0.01 | 0.01 | 0.01 | 0.01 | 0.01 | 0.01 |
| Hyperparameters w.r.t. the SGC model | | | | | | |
| # Layers | 1 | 1 | 1 | N.A. | N.A. | N.A. |
| Hidden Dimension | N.A. | N.A. | N.A. | N.A. | N.A. | N.A. |
| Dropout | N.A. | N.A. | N.A. | N.A. | N.A. | N.A. |
| Optimizer | AdamW with $1e - 5$ weight decay | | | | | |
| Pretrain Steps | 300 | 500 | 500 | N.A. | N.A. | N.A. |
| Training Steps | 15 | 40 | 30 | N.A. | N.A. | N.A. |
| Learning Rate | 0.01 | 0.01 | 0.01 | N.A. | N.A. | N.A. |
| Hyperparameters w.r.t. the GAE model | | | | | | |
| # Layers | N.A. | N.A. | N.A. | 2 | 2 | 2 |
| Hidden Dimension | N.A. | N.A. | N.A. | $[32, 16]$ | $[32, 16]$ | $[32, 16]$ |
| Activation | N.A. | N.A. | N.A. | N.A. | N.A. | N.A. |
| Dropout | 0.0 | 0.0 | 0.0 | 0.0 | 0.0 | 0.0 |
| Optimizer | AdamW with $1e - 5$ weight decay | | | | | |
| Pretrain Steps | N.A. | N.A. | N.A. | 200 | 200 | 200 |
| Training Steps | N.A. | N.A. | N.A. | 60 | 100 | 50 |
| Learning Rate | N.A. | N.A. | N.A. | 0.01 | 0.01 | 0.01 |

## E   DATASET DESCRIPTIONS

We provide additional details about the datasets utilized in our work, as discussed in § 4.1:

- Citation Networks: Each node corresponds to a paper, and an edge between two nodes represents the citation relationship between the respective papers. The node attributes in these networks are generated using the bag-of-words model based on the abstract sections of the published papers.

- Co-author Networks: It consist of nodes representing authors, where an edge between two nodes indicates that the corresponding authors have collaborated on a paper. The node attributes in these networks are constructed based on the bag-of-words model using the authors' profiles.

- Social Networks: Each node represents a user, and the links between nodes represent interactions between users. The attributes associated with these nodes are derived from user profiles.

The datasets used in our work are referred to as CS and Phy, which are abbreviations for the Co-author-CS and Co-author-Phy datasets, respectively. A comprehensive overview of the datasets, including their detailed statistics, is presented in Table 10.

Table 10: Detailed statistics of the datasets used for node classification and link prediction. We follow the default settings of Redress (Dong et al., 2021) fair comparisons.

| Task | Dataset | # Nodes | # Edges | # Features | # Classes |
|---|---|---|---|---|---|
| node classification | ACM | $16,484$ | $71,980$ | $8,337$ | 9 |
| | Coauthor-CS | $18,333$ | $81,894$ | $6,805$ | 15 |
| | Coauthor-Phy | $34,493$ | $247,962$ | $8,415$ | 5 |
| link prediction | BlogCatalog | $5,196$ | $171,743$ | $8,189$ | N.A. |
| | Flickr | $7,575$ | $239,738$ | $12,047$ | N.A. |
| | Facebook | $4,039$ | $88,234$ | $1,406$ | N.A. |

# F    RANKING-BASED INDIVIDUAL FAIRNESS IN REAL WORLDS

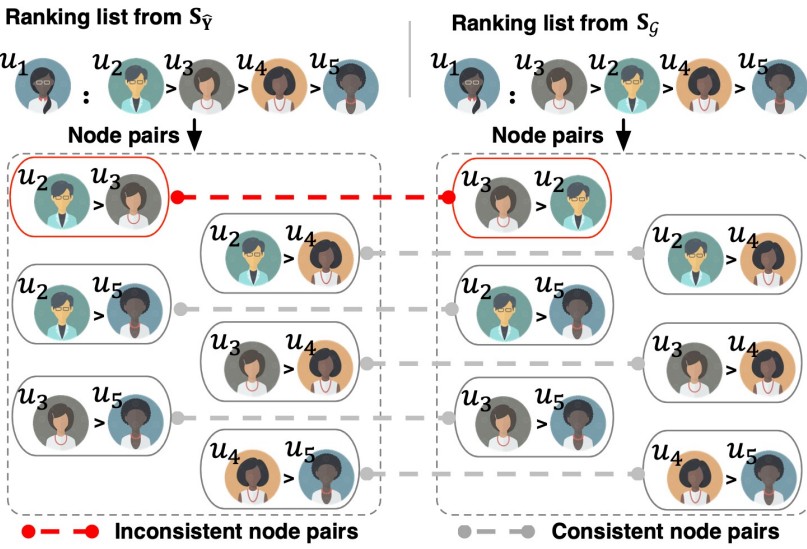

Figure 2: An example figure motivating individual fairness in graphs; **taken from** Dong et al. (2021).

**Motivating Example**    Consider a job recommendation platform utilizing a GNN to rank candidates for employers. In this scenario, nodes symbolize individuals, while edges represent connections or similarities (such as shared skills, experience, or educational background). Figure 2, by Dong et al. (2021) as our motivating example, showcases two distinct ranking lists for different candidate subsets, $S_y$ and $S_g$, based on their profiles and network connections.

In an ideally fair system, candidates possessing comparable qualifications and experiences should receive consistent rankings across various subsets. However, our example reveals disparities (indicated by red dotted lines) that arise without the implementation of our proposed framework. These disparities manifest as inconsistent rankings for similar candidates, resulting in potential unfairness. For instance, the ranking of candidate $u_2$ is higher than $u_4$ in the list from $S_y$, but this order reverses in the list from $S_g$. Such inconsistency can lead to overlooking qualified candidates based on the subset they belong to, challenging the notion of individual fairness.

Our framework employs the Lipschitz condition to establish a boundary, ensuring that the output differences (rankings) for any pair of similar candidates remain within the limits set by their feature-space distances. This mechanism acts as a safeguard, preserving ranking consistency across different scenarios and thereby maintaining individual fairness.

**Practical Value**    The real-world significance of our approach is its capacity to generate fair and consistent outcomes in various applications where GNNs are used for critical decision-making. This includes domains like credit scoring, social media content ranking, and personalized medicine, where inconsistent results can significantly affect individuals' opportunities and well-being. Our objective is to cultivate fairness and trust in AI systems, ensuring they adhere to societal and ethical standards.

In conclusion, our approach is not only theoretically robust but also holds substantial practical merit. It provides a significant benefit to both individuals impacted by GNN-based decisions and the organizations implementing these models. We believe the provided motivating example effectively demonstrates the potential influence and real-world relevance of our research.

