# OpenReview forum: "Aligning Relational Learning with Lipschitz Fairness"
_ICLR.cc/2024/Conference — ICLR 2024 poster_

### Official Review · Reviewer_kSfC · 2023-10-29

**Soundness:** 3 good
**Presentation:** 3 good
**Contribution:** 2 fair
**Rating:** 5
**Confidence:** 1

**Summary:**

I’m not an expert in the GNN, fairness community but has some experience on the estimation of Lipschitz constant.
Accordingly, my reviews are purely based on reading the paper without comparison to the related literature on GNN and fairness.

In my understanding, this paper studies the use of Lipschitz constant of graph neural networks for fairness. To derive the Lipschitz constant, the Jacobian based approach is proposed and verified by several experiments on classification and prediction tasks.

**Strengths:**

- Estimate the Lipschitz constant of GNNs via Jacobian matrix
- Comprehensive experiments on GNNs for the bias

**Weaknesses:**

Since I’m not familiar with GNN and fairness, it appears difficult to evaluate the contribution.

For the estimation of the Lipchitz constant, the approach is based on the Jacobian matrix, which is loose. In this community, there are several advanced and promising approaches for the Lipschitz constant estimation on DNNs. So I’m wondering how this can be extended to the GNN setting, and what’s the difficulty issue behind this?

**Questions:**

See the above.

---

> ### Author Response · Authors · 2023-11-19
> **Response to Reviewer kSfC (1/2)**
>
> We are grateful for Reviewer kSfC's insightful feedback and appreciate the opportunity to address your queries. Your perspectives, especially regarding the estimation of the Lipschitz constant, add valuable context to our discussion. We have incorporated additional background information and clarified the challenges associated with our approach in the updated manuscript for enhanced understanding.
>
> ___
>
>
> **[Q1: GNN and Fairness -- A simplified, easy-to-understand overview]**
>
> Thank you for your comment and the opportunity to elucidate the concepts of Graph Neural Networks (GNNs) and individual fairness that are central to our work. We appreciate that these are specialized areas, and we aim to provide a clear and concise explanation.
>
> **Graph Neural Networks (GNNs):**
>
> GNNs are designed to process graph or relational data. Unlike conventional data structures, graphs consist of nodes and edges, which represent entities and their relationships. GNNs excel in capturing complex patterns in such data by aggregating information from neighboring nodes, making them useful for tasks like classification, prediction, and clustering.
>
>
> **Ranking-based Individual Fairness:**
>
> The concept of individual fairness in the context of ranking is centered around the principle that individuals who are similar, based on defined attributes, should be accorded comparable rankings. This principle is particularly pertinent in the domain of GNNs, where the objective often involves ranking nodes — such as users, items, or entities — based on a blend of their intrinsic features and the nature of their interconnections within the graph.
>
> In this approach, the focus is on the relative positioning of individuals or nodes within the ranking order. The key is not the absolute scores or ratings assigned to each individual, but rather ensuring that the rank order is fair and equitable. This means that if two nodes are deemed similar in terms of their attributes or their roles within the network, this similarity should be reflected in how they are ranked. Whether it's in generating search results, recommendations, or categorizing items, the ranking-based fairness approach strives to preserve the integrity of this relative ordering, maintaining consistency across various representations or subsets of the data. It's about upholding a fair and justifiable hierarchy that resonates with the underlying similarity and relationship patterns among the nodes in the graph.
>
>
> **Lipschitz Condition and Individual Fairness (in our work):**
>
> The Lipschitz condition is a mathematical concept that, in this context, can be used to enforce a form of individual fairness. A function is said to be Lipschitz continuous if there exists a constant $L$ such that for any two points $x$ and $y$, the difference in the function's outputs is at most $L$ times the distance between $x$ and $y$. In simpler terms, similar inputs lead to similar outputs with a constant on how different the outputs can be.
>
> When applied to ranking, a Lipschitz condition can ensure that if two nodes are similar (close in the graph structure or feature space), their differences in ranking (output of the GNN) are limited. This prevents wildly different rankings for similar nodes, which contributes to individual fairness.
>
>
> **To enhance the accessibility of our paper for a broader machine learning audience, we have included a clear and concise overview of GNNs and individual fairness in Appendix A.2. We are grateful to the reviewer for their valuable input, which guided us in making our work more understandable to readers from diverse backgrounds in the field of machine learning.**
>
> ___

---

> ### Author Response · Authors · 2023-11-19
> **Response to Reviewer kSfC (2/2)**
>
> ___
> **[Q2: How this can be extended to the GNN setting, and what’s the difficulty issue behind this?]**
>
> ***Extending Lipschitz constant estimation methods from traditional DNNs, like MLPs or CNNs to GNNs presents several challenges:***
>
> 1. _Limited Transferability:_ The methods used for calculating Lipschitz constants in DNNs are not always directly applicable to GNNs. This limited transferability means that only a subset of these methods can be adapted for use with GNNs. Research in the DNN domain often focuses on simpler architectures like MLPs, as seen in studies [1, 2, 3], or specific components like self-attention layers [4], or even reformulates the problem into a numerical optimization challenge [5]. However, these approaches are tailored to specific models or layers and do not generalize well to the more intricate structures of GNNs, making them less suitable for direct application in the GNN context.  Furthermore, the computational cost associated with MLPs makes it challenging to extend to the more complex GNNs.
>
> 2. _Complexity Challenges in GNNs:_ The introduction of graph topological structures in GNNs results in layers that are inherently more complex than those in traditional DNNs. This added complexity stems from the need to integrate the graph's topology with the weight matrix, a consideration not present in standard DNN frameworks. Existing studies [1, 2, 3, 4, 5], which primarily focus on single-layer feedforward neural networks, do not take these topological aspects into account, limiting their direct applicability to GNNs.
>
> As a consequence, while these methods are effective in achieving tight Lipschitz constants for simpler models like MLPs, CNNs, or statistical models, their suitability for GNNs is constrained. The methods fall short in terms of transferability and introduce excessive complexity. This makes them less practical for implementing real-time regularization in GNNs during the training process.
>
>
> ***Advantages and contributions of the proposed JacoLip method:***
>
> 1. _Generalization:_  Our JacoLip method is a versatile framework designed for computing Lipschitz constants across various GNN models, including GCN, SGC, and GAE. This demonstrates its broad adaptability and suitability for different GNN architectures.
>
> 2. _Practicality:_ While the Lipschitz constant derived using the JacoLip method, based on the Jacobian matrix, is not the tightest, it is designed to be computationally efficient and well-integrated with the GNN training process. This approach emphasizes practical applicability and dynamic adjustment during training to balance individual fairness with performance.
>
> 3. _Efficiency:_ JacoLip efficiently computes the Lipschitz constant by utilizing the gradient computation capabilities of the PyTorch framework. This enables the method to calculate the constant without adding a significant computational burden.
>
> In essence, the JacoLip method, while not providing the tightest bound on the Lipschitz constant, offers a solution that is practical, efficient, and adaptable to the unique characteristics and training requirements of GNNs.
>
>
> ___
>
> **References:**
>
> [1] Spectral Norm Regularization for Improving the Generalizability of Deep Learning, Arxiv 2017
>
> [2] Regularisation of Neural Networks by Enforcing Lipschitz Continuity, Machine Learning (springer) 2021
>
> [3] Lipschitz Regularity of Deep Neural Networks: Analysis and Efficient Estimation, NeurIPS 2018
>
> [4] The Lipschitz Constant of Self-Attention, ICML 2021
>
> [5] Efficient and Accurate Estimation of Lipschitz Constants for Deep Neural Networks, NeurIPS 2019

---

> ### Author Response · Authors · 2023-11-22
>
> We hope that our response has helped explain our work's contributions. Please feel free to let us know if you have any further questions.

---

### Official Review · Reviewer_ANH8 · 2023-10-31

**Soundness:** 3 good
**Presentation:** 3 good
**Contribution:** 3 good
**Rating:** 8
**Confidence:** 3

**Summary:**

This paper designs an approach to estimate the Lipschitz constant for GNNs efficiently. Adding such an estimation as a regularization to limit output changes induced by input biases helps align the model with principles of rank-based individual fairness. The authors validate the proposed method through experiments on node classification and link prediction tasks. Results show the approach effectively improves fairness while maintaining accuracy. In particular, this paper provides insights into how constraining the Lipschitz constant influences training dynamics and the trade-off between accuracy and fairness.

**Strengths:**

(1) This paper is generally well-organized and easy to follow. Plus, a comprehensive theoretical analysis is provided.

(2) Extensive experiments are performed and presented. The superiority exhibited by the experimental results in most cases seems promising.

(3) The efficient computation of the Lipschitz constant via the Jacobian matrix makes the proposed approach scalable.

**Weaknesses:**

(1) This paper lacks a motivating example to deliver the significance of the proposed approach in applications.

(2) No time complexity / running time comparison is provided. This undermines the claimed advantage in efficiency.

(3) While experiments validate the effectiveness of the proposed approach, more analysis could be provided on the sensitivity of the method to different hyperparameters.

**Questions:**

(1) What is the motivation of the proposed approach? The practical value would be much clearer if any motivating example could be provided.

(2) What is the time complexity of the proposed approach? It would be desired if any time complexity analysis / running time comparison could be provided.

(3) Is the proposed framework sensitive to the value of $u$ (i.e., the weight assigned for the regularization term) in Algorithm 1?

---

> ### Author Response · Authors · 2023-11-19
> **Response to Reviewer ANH8 (1/3)**
>
> We thank the Reviewer ANH8, for your insightful review and comments. We have addressed your comments below, and these updates are also reflected in the revised version of our paper.
>
>
> ___
>
>
> **[Q1&W1. What is the motivation of the proposed approach? A motivating example to deliver the significance of the proposed approach in applications is helpful.]**
>
> We appreciate your constructive feedback on the need to illustrate the practical applications of our ranking-based individual fairness approach. Recognizing the importance of this aspect, we have added a motivating example, including a figure in Appendix F of our revised manuscript. Below we summarize this example (please refer to the revised paper to see the figure).
>
>
> **Motivating Example:**
> Consider a job recommendation platform utilizing a GNN to rank candidates for employers. In this scenario, nodes symbolize individuals, while edges represent connections or similarities (such as shared skills, experience, or educational background). The figure in our motivating example showcases two distinct ranking lists for different candidate subsets, $S_y$ and $S_g$, based on their profiles and network connections.
>
> In an ideally fair system, candidates possessing comparable qualifications and experiences should receive consistent rankings across various subsets. However, our example reveals disparities (indicated by red dotted lines) that arise without the implementation of our proposed framework. These disparities manifest as inconsistent rankings for similar candidates, resulting in potential unfairness. For instance, the ranking of candidate $u_2$ is higher than $u_4$ in the list from $S_y$, but this order reverses in the list from $S_g$. Such inconsistency can lead to overlooking qualified candidates based on the subset they belong to, challenging the notion of individual fairness.
>
> Our framework employs the Lipschitz condition to establish a boundary, ensuring that the output differences (rankings) for any pair of similar candidates remain within the limits set by their feature-space distances. This mechanism acts as a safeguard, preserving ranking consistency across different scenarios and thereby maintaining individual fairness.
>
> **Practical Value:**
> The real-world significance of our approach is its capacity to generate fair and consistent outcomes in various applications where GNNs are used for critical decision-making. This includes domains like credit scoring, social media content ranking, and personalized medicine, where inconsistent results can significantly affect individuals' opportunities and well-being. Our objective is to cultivate fairness and trust in AI systems, ensuring they adhere to societal and ethical standards.
>
> In conclusion, our approach is not only theoretically robust but also holds substantial practical merit. It provides a significant benefit to both individuals impacted by GNN-based decisions and the organizations implementing these models. We believe the provided motivating example effectively demonstrates the potential influence and real-world relevance of our research.
>
> **To facilitate a better intuitive understanding of the problem of individual fairness, we have incorporated this motivating example in Appendix F of our revised manuscript.**
>
> ___

---

> ### Author Response · Authors · 2023-11-19
> **Response to Reviewer ANH8 (2/3)**
>
> ___
>
> **[Q2&W2. What is the time complexity of the proposed Lipschitz constant estimation approach? It would be desired if any time complexity analysis / running time comparison could be provided.]**
> The time complexity of our Lipschitz constant estimation is primarily governed by the computation of gradients and their norms. The code computes the Lipschitz constant without incurring the computational overhead typically associated with second-order derivatives.
> The relevant portion of our code (anonymously at: https://tinyurl.com/JacoLip) is:
> ```python
> for i in range(out.shape[1]):
>     # Directional derivative vectors are initialized as zero vectors
>     v = torch.zeros_like(out)
>     v[:, i] = 1
>
>     # Compute gradients with respect to the inputs, not the model parameters
>     gradients = autograd.grad(outputs=out, inputs=input, grad_outputs=v,
>                               create_graph=True, retain_graph=True, only_inputs=True)[0]
>
>     # We only compute the norm of these gradients, which is a first-order operation
>     grad_norm = torch.norm(gradients, dim=1).unsqueeze(dim=1)
>     lip_mat.append(grad_norm)
> ```
> The time complexity for each iteration is determined by the `autograd.grad` function, which computes the gradients with respect to the inputs. The complexity of gradient computation is typically $O(NM)$, where $N$ is the number of elements in the input tensor in the current batch, and $M$ is the number of classes (thanks to the nature of relational datasets, the model output corresponding to their node feature usually has a small dimension. For example, $M$ on ACM is $9$, $M$ on Coauthor-CS is $15$, $M$ on Coauthor-Phy is $5$: these are small constants).
>
> However, since we iterate over the second dimension of the output tensor (`out.shape[1]`), which corresponds to the number of classes (small constant too), the loop's total complexity becomes $O(NM^2)$.
>
> Subsequently, the norm calculation is within the loop, it does not change the overall complexity of $O(NM^2)$ for the Lipschitz constant estimation part.
>
>
> Regarding the empirical running time, our experiments (conducted for this rebuttal) show that the additional computational load required for estimating the Lipschitz constant is minimal, especially when compared to the overall training process, including standard training. This efficiency is largely due to the advanced gradient computation capabilities inherent in PyTorch's autograd system, coupled with our strategy to bypass the computation of the full Jacobian matrix.
>
> To demonstrate the efficiency of our method, we utilized the FB15k-237 dataset as a benchmark. This dataset consists of 4,039 nodes, 88,234 edges, and 1,406 attributes, making it a suitable candidate for evaluating our approach. In the accompanying table, we compare the time and GPU memory usage of various methods when applied to the FB15k-237 dataset. These comparative results clearly illustrate that our method does not introduce significant computational overhead when compared to the standard training procedure.
>
> Table B. Comparative analysis of time and GPU memory usage for various methods applied to the FB15k-237 dataset.
>
> |                 Method                  | Lipschitz Tightness | Graph Structure | Efficiency | Time per Iter (FB dataset) | GPU Memory-Usage (FB dataset) |
> |:---------------------------------------:|:-------------------:|:---------------:| :---: |:--------------------------:|:-----------------------------:|
> |             GCN (standard)              |        N.A.         |      N.A.       | N.A. |          0.2841s           |           414.8 MiB           |
> |  Lipschitz originally for CNN/MLP [1]   |     Restrictive     |       No        | No        |          3.2407s           |          2130.1 MiB           |
> | Layered Lipschitz Computing for GNN [2] |     Restrictive     |       Yes       | No        |          3.8541s          |          3280.7 MiB           |
> |          Ours (GCN + JacoLip)           |     Restrictive     |       Yes       | Yes       |          0.2972s           |           420.0 MiB           |
>
> References:
>
> [1] Efficiently Computing Local Lipschitz Constants of Neural Networks via Bound Propagation, NeurIPS 2022
>
> [2] Enhancing Node-Level Adversarial Defenses by Lipschitz Regularization of Graph Neural Networks, KDD 2023
>
> **We include this analysis in Appendix C.4 and C.5 of our revised paper.**
>
> ___

---

> ### Author Response · Authors · 2023-11-19
> **Response to Reviewer ANH8 (3/3)**
>
> ___
>
> **[Q3&W3. While experiments validate the effectiveness of the proposed approach, more analysis (i.e., $\mu$ the weight assigned for the regularization term) could be provided on the sensitivity of the method to different hyperparameters.]**
>
>  We acknowledge the importance of understanding how varying hyperparameters impact the Lipschitz constant of the GNN, and consequently, the balance between individual fairness and performance. To provide a comprehensive analysis, we experimented with different values of the hyperparameter $\mu$, which directly influences the Lipschitz constant of the model. A larger value of $\mu$ correlates with a smaller Lipschitz constant, making it a pivotal factor in our approach. Below are two tables illustrating the effects of varying $\mu$ on the GCN model applied to the Node Classification task on the Co-author-CS dataset. These tables present the outcomes in terms of AUC and NDCG metrics:
>
> Table C. GCN on Co-author-CS dataset with varying $\mu$ based on Feature Similarity
>
> | u        | 0 | 0.1   | 0.01  | 0.001 | 0.0001 | 0.00001| 0.000001|
> |----------| ---- | ----  |  ---- | ----  | ----   | ----   |----     |
> | NDCG@10↑ | 50.84 |37.67| 48.98  | 63.56 |  68.32 |  67.91 |  66.26 |
> | AUC↑   | 90.59| 60.18 | 89.96|  90.96|   90.20|   90.23|   90.18|
>
> Table D. GCN on Co-author-CS dataset with varying $\mu$ based on Structure Similarity
>
> | u        | 0 | 0.1   | 0.01  | 0.001 | 0.0001 | 0.00001|0.000001|
> |----------| ----- |----  |  ---- | ----  | ----   | ----   | ----   |
> | NDCG@10↑ |18.29| 4.11  | 10.66 | 19.62  | 31.82 | 26.82 | 19.42   |
> | AUC↑  |90.34 |33.46 | 86.98 | 89.25  | 89.12 | 89.18 | 89.51   |
>
>
> The results from these tables clearly show that both AUC and NDCG metrics are sensitive to variations in the Lipschitz constant, which is controlled by $\mu$. Notably, when $\mu$ is relatively large, resulting in a smaller Lipschitz constant, there is a significant impact on both performance and fairness metrics. As $\mu$ decreases, allowing the Lipschitz constant of the model to approach that of the original model, we observe a convergence of both performance and fairness metrics towards those of the original model.
>
> This analysis highlights the delicate balance that must be maintained between fairness and performance in our approach. The data reveals that the metric of fairness is more responsive to changes in $\mu$ compared to performance. Therefore, identifying an optimal Lipschitz constant that adequately meets both fairness and performance criteria becomes paramount in our methodology.
>
> **We have included this detailed hyperparameter sensitivity analysis in Appendix C.3 of our revised manuscript.**
> ___

---

> ### Author Response · Authors · 2023-11-22
>
> We hope that our response has helped explain our work's contributions. Please feel free to let us know if you have any further questions.

---

> ### Comment · Reviewer_ANH8 · 2023-11-23
> **Thanks for the response**
>
> Thanks for the detailed response from the reviewer. Most of my concerns have been addressed and I had my score raised.

---

> > ### Author Response · Authors · 2023-11-23
> > **Thank you!**
> >
> > We sincerely thank you again for your valuable feedback. It is encouraging to know that our detailed revisions have addressed your concerns. Your acknowledgment of our efforts to improve our work is greatly appreciated!

---

### Official Review · Reviewer_uCQA · 2023-11-01

**Soundness:** 2 fair
**Presentation:** 2 fair
**Contribution:** 2 fair
**Rating:** 6
**Confidence:** 3

**Summary:**

The paper focus on individual fairness in GNNs by constraining the output perturbations induced by input biases. The authors propose a Lipschitz constant-based approach to examine the output stability of GNNs and demonstrate its effectiveness in limiting biases in the model output. They also introduce a computational strategy using the Jacobian matrix to efficiently compute the Lipschitz constant. The paper includes experiments on real-world datasets and comparisons with existing methods.

**Strengths:**

1. The paper addresses an important and timely problem of fairness in GNNs, which has gained significant attention in recent years.

2. The use of Lipschitz constants to control the stability of GNN outputs provides a provable method for examining output stability without additional computational costs.

3. The theoretical analysis and formulation of the Lipschitz constant for GNNs operating on graph data is well-structured.

**Weaknesses:**

1. The proposed method does not need manual annotation of sensitive attributes. However, lack of information would surely compromise the guarantee of ensuring fairness. Discussions on the drawback compared to other fairness methods that explicitly use sensitive attributes should be included.

2. The methodology in sec 3 does not appear to be specifically tailored for fairness problems. Is there any guarantee on how fairness regarding sensitive attributes is ensured for the proposed JacoLip algorithm?

3. The experimental results are not organized clearly. For example, in Table 1 JacoLip isn’t always achieving the best performance, but best results are not in bold or otherwise marked, making it difficult to compare across methods and datasets.

**Questions:**

See weaknesses

---

> ### Author Response · Authors · 2023-11-19
> **Response to Reviewer uCQA (1/3)**
>
> We would like to thank Reviewer uCQA for your thoughtful comments and questions regarding our submission. Here we address your concerns in the following.
>
> We also integrate the updates into the revised manuscript for enhancing clarity, and substantiate the claims and contributions of our paper.
>
> ___
>
> **[1. Discussions on the drawback compared to other fairness methods that explicitly use sensitive attributes should be included.]**
>
> We appreciate the opportunity to clarify the methodological choices made in our research, particularly concerning the non-reliance on manual annotation of sensitive attributes.
>
> Our work focuses on individual fairness, which seeks to ensure that similar individuals are treated similarly, independent of their membership in any demographic group. This approach inherently **differs** from the methods for group fairness, which requires knowledge of sensitive attributes to ensure equal treatment across predefined groups.
>
> The confusion may arise from the background setting of group fairness, which indeed necessitates manual annotation of sensitive attributes to assess and guarantee fairness across groups. However, individual fairness, the focus of our method and the related baselines, operates on a different premise: it does not require, nor does it benefit from, the manual annotation of sensitive attributes because it does not assess fairness on a group level. Instead, our method evaluates the fairness of outcomes through a metric of individual similarity, ensuring that the decision-making process is equitable on a case-by-case basis.
>
> To explicitly address the concern raised, the lack of sensitive attribute annotation in the context of individual fairness is **not a drawback but a feature**. It aligns with the goal of providing fairness without the need to define or reference demographic groups, thus sidestepping the risks of reinforcing stereotypes or biases associated with these groups. This methodological choice is also in line with privacy preservation and the ethical standpoint of minimizing the use of potentially/hidden sensitive personal information.
>
> We would like to emphasize that the datasets and those used by baselines for individual fairness do not include sensitive attribute annotations precisely because our approach, as well as those compared against, are designed to uphold individual fairness. This design choice might not be seen as a compromise in ensuring fairness but rather as an adherence to the principles of individual fairness, which does not rely on group definitions (e.g., sensitive attributes). We believe this clarification underscores the value and novelty of our contribution to graph individual fairness.
>
> **We include this discussion in Appendix A.1 of our revised paper, as quoted below:**
> > The concepts of individual and group fairness are fundamental in the domain of machine learning ethics, particularly when designing and evaluating algorithms for fair decision-making. Both concepts aim to address fairness concerns, but they do so from different perspectives and with distinct implications. While group fairness relies on annotated sensitive attributes, individual fairness does not. The distinction between individual and group fairness in terms of reliance on annotated attributes highlights a fundamental difference in how these fairness paradigms conceptualize and address fairness.
> >
> > Individual Fairness: This concept is centered around the notion of treating similar individuals similarly. In a machine learning context, it implies that if two individuals are similar with respect to the attributes relevant to a decision-making process (e.g., loan approval, job recruitment), they should be treated in a comparable manner by the algorithm.  Operationalizing individual fairness often involves defining a suitable metric of similarity between individuals, which can be challenging. This metric should capture all the relevant aspects that justify similar treatment. One of the main challenges is the subjective nature of defining similarity. What constitutes "similar" in one context or for one set of stakeholders might not be agreed upon universally. There's also a computational challenge in ensuring this kind of fairness at scale, as it may require pairwise comparisons among individuals.

---

> ### Author Response · Authors · 2023-11-19
> **Response to Reviewer uCQA (2/3)**
>
> > Individual fairness has (1) No Need for Annotated Attributes: Individual fairness typically does not rely on explicitly annotated attributes, especially sensitive attributes like race, gender, or age. Instead, it focuses on the idea of treating similar individuals similarly, where similarity is often defined in the context of the specific task or decision process. Instead, it relies on (2) Implicit Attributes: The concept of similarity in individual fairness is based on implicit attributes derived from the context or the nature of the task. These attributes are usually not explicitly labeled but are inferred from the data or the specific decision-making scenario.
> >
> > Group Fairness: This approach focuses on ensuring fairness across predefined groups, typically defined by explicit sensitive attributes like race, gender, or age. Group fairness is concerned with statistical measures and often aims for equal treatment or outcomes across these groups. Common criteria include demographic parity, equal opportunity, and equalized odds. Group fairness is easier to quantify and implement than individual fairness as it relies on statistical measures (e.g., ensuring that selection rates for a job are equal across different gender groups). Group fairness is often applied in large-scale decision-making scenarios where societal or policy-level fairness concerns are paramount, such as in credit scoring or hiring processes. A significant challenge with group fairness is the risk of oversimplification. By focusing on broad groups, it might overlook nuances and individual-level disparities within these groups. Additionally, it can sometimes lead to unfair outcomes for individuals when trying to balance statistics at the group level.
> >
> > Group fairness (1) Relies on Annotated Attributes: Group fairness explicitly relies on annotated attributes, often focusing on sensitive or protected attributes. These are explicit labels in the dataset, such as race, gender, or other demographic information. It uses (2) Explicit Categories: In group fairness, individuals are categorized based on these explicit attributes, and fairness is measured by evaluating the outcomes or treatments across these predefined groups. This approach simplifies the fairness problem by reducing it to a series of statistical measures (like demographic parity, equal opportunity, etc.) across these groups. While this simplification aids in quantification and implementation, it may overlook individual-level disparities and nuanced differences within groups.
>
>
> ___

---

> ### Author Response · Authors · 2023-11-19
> **Response to Reviewer uCQA (3/3)**
>
> ___
>
> **[2. The methodology in sec 3 does not appear to be specifically tailored for fairness problems. Is there any guarantee on how fairness regarding sensitive attributes is ensured for the proposed JacoLip algorithm?]**
>
> Thank you for your question about the application of our JacoLip algorithm to fairness problems. We affirm that JacoLip is indeed tailored to address individual fairness in GNNs. This is accomplished by carefully evaluating and processing nodes' features and their connectivity patterns within the graph. Specifically, JacoLip aims to minimize the impact of biased features, thereby ensuring consistent treatment of similar individuals.
>
> To provide a more detailed explanation, we elaborate on the role of Lipschitz constants in our JacoLip algorithm and how they contribute to ensuring individual fairness:
>
> ### How Lipschitz constants can guarantee individual fairness (relationship between Lipschitz constant and individual fairness)
> In the context of individual fairness on graphs, the relationship between ranking-based individual fairness and Lipschitz continuity is established through Lipschitz constraints on the GNN model. The Lipschitz continuity ensures that the model's predictions are not overly sensitive to small changes in node embeddings. Let's define the two concepts and their relationship mathematically:
> - **Lipschitz Property:** In the GNN model $f : V \rightarrow \mathbb{R}^d$, Lipschitz continuity constrains the change in model output concerning small **changes/biases** in input (node embeddings). A function $f$ is Lipschitz continuous if there exists a constant $K$ such that $\|f(z_i) - f(z_j)\| \leq K \|z_i - z_j\|$ for any two input points $z_i$ and $z_j$. Here, $\| \cdot \|$ denotes a norm (e.g., $L2$ norm), and $K$ is the Lipschitz constant.
>
> - **Ranking-based Individual Fairness:** Based on prior work [1,2], this concept in GNNs focuses on maintaining the rank-ordering of pairwise node **similarities** pre and post-training. This is achieved by a ranking loss function $L(\cdot)$ that penalizes discrepancies between predicted pairwise distances and original similarities as $\text{Ranking Loss: } \mathcal{L}\_{\text{rank}} = \sum_{(v\_i, v\_j) \in \mathcal{P}} L(S(v\_i, v\_j), D(z\_i, z\_j))$, where $\mathcal{P}$ is the set of all node pairs, and $L(\cdot)$ measures the discrepancy between predicted pairwise distance $D(z_i, z_j)$ and original similarity $S(v_i, v_j)$.
>
> - **The Exact Relationship:**  By imposing a Lipschitz constraint on the GNN, the model’s output becomes less sensitive to minor variations (i.e., **minor biases** ) in node embeddings, thereby reinforcing ranking-based individual fairness.  The mathematical connection can be expressed as $\text{RankLoss}(S(v_i, v_j), D(z_i, z_j)) \propto L(f(z_i), f(z_j))$, where $\text{RankLoss}(S(v_i, v_j), D(z_i, z_j))$ is the ranking loss function, measuring the discrepancy between predicted pairwise distances $D(z_i, z_j)$ and original similarities $S(v_i, v_j)$ in the output space.
>
> Enforcing a Lipschitz constraint on the GNN's model parameters or architecture inherently promotes the preservation of the rank-ordering of node similarities during training. This approach drives the GNN to uphold individual fairness by ensuring that nodes with similar attributes receive comparable rankings in the output space, resulting in fair and unbiased predictions for each node.
>
> In summary, the principle of Lipschitz continuity in GNNs is directly related to the concept of ranking-based individual fairness. It achieves this by moderating the model's response to small variations in the embeddings of nodes.
>
> We have incorporated this comprehensive discussion on shaping Lipschitz constants to assure individual fairness into Appendix B.5 of our revised manuscript.
>
>
> ___
>
>
> **[3. In Table 1 JacoLip isn’t always achieving the best performance, but best results are not in bold or otherwise marked, making it difficult to compare across methods and datasets.]**
>
> Thank you for your helpful feedback. We have revised Table 1 to boldface the best-performing results for each dataset, enhancing comparability across methods and datasets. Similarly, Table 2 has been updated to clearly indicate the top results.
>
> ___
>
> **References:**
>
> [1] Individual fairness for graph neural networks: A ranking based approach, KDD 2021
>
> [2] InFoRM: Individual Fairness on Graph Mining, KDD 2020

---

> > ### Comment · Reviewer_uCQA · 2023-11-21
> > **Thanks for the rebuttal**
> >
> > Thank you for your comprehensive response. Most of my concern has been addressed. I have increased my score.

---

> > > ### Author Response · Authors · 2023-11-22
> > > **Thank you!**
> > >
> > > Thank you again for your insightful feedback. We are glad to hear that our comprehensive revisions have addressed your main concerns. We appreciate you recognizing our efforts in improving our work.

---

### Official Review · Reviewer_7Erc · 2023-11-03

**Soundness:** 3 good
**Presentation:** 2 fair
**Contribution:** 3 good
**Rating:** 6
**Confidence:** 3

**Summary:**

This paper addresses the problem of individual fairness in the relational learning task (graph datasets). The measure of fairness in this work fairness is based on the notion that similar inputs must have similar outputs. The authors argue that to ensure fairness, Lipschitz's constant of GNN must be small. To this end, they characterize the lipschitz's constant in Jacobian and optimize it along with the loss function to achieve fairness. Empirical results show improved individual fairness at competitive or better utility.

**Strengths:**

- Characterizing fairness in terms of the Lipschitz constant and optimizing this constant is an interesting approach to achieving individual fairness in GNNs.
- The proposed approach, JacoLip, outperforms other fair learning methods --- JacoLip achieves a better fairness score while having the best or close to the best utility. Further, the authors show that it can even improve an existing fair learning method to some extent. These observations, coupled with the evaluation over several datasets, increase confidence in the proposed idea.

**Weaknesses:**

### Computational Efficiency
- At several places in the paper, the authors have emphasized that their method has no computational overhead. However, no empirical evidence is provided for this. For example, comparing training time and memory usage of different approaches could be a way to support this argument.
- The proposed approach involves computing norms of the Jacobian matrix in the loss function. This could lead to computing gradient of gradient during backpropagation (implicitly by PyTorch), which can be memory and compute-intensive. How the authors got around this is unclear.

**Questions:**

- In the introduction, the authors mention, "... computational approach utilizes intermediate tensors extracted..." What intermediate tensors are being referred to here?
- Eq 7 describes the jacobian of the i-th node. Shouldn't it be a 3D tensor? $Y_{jk}$ (output of node-j) depends on $X_i$ (node i) due to message passing. Are $\frac{dY_{j1}}{dX_{i1}}$ $i\neq j$ and such terms ignored? I think the discussion about the computation of the Lipschitz constant could be more elaborate, with a clear indication of the dimension of the tensors.

I think the paper shows good empirical results. Still, it would help to make the section about the Lipschitz constant more straightforward and compare training time and memory usage to support computational efficiency claims.

---

> ### Author Response · Authors · 2023-11-19
> **Response to Reviewer 7Erc (1/3)**
>
> We would like to thank Review 7Erc's kind acknowledgment and affirmation. Here we address your comments in the following. We also integrate our answers into our updated manuscript for better clarity.
> ___
>
> **[W1. No empirical evidence is provided for "no computational overhead" claim]**
>
> To address the concern raised, we provide a detailed comparison using the FB15k-237 dataset. This dataset comprises 4,039 nodes, 88,234 edges, and 1,406 attributes, making it a representative choice for our analysis. The table below presents a comparative analysis of time and GPU memory usage for various methods applied to the FB15k-237 dataset. The results from this comparison demonstrate that our proposed method incurs no additional computational overhead when contrasted with the standard training procedures. These findings offer a more concrete empirical evidence to support it. **We have included this analysis in Appendix C.4 of our revised paper.**
>
> Table A. Comparative analysis of time and GPU memory usage for various methods applied to the FB15k-237 dataset.
>
> |                 Method                  | Lipschitz Tightness | Graph Structure | Efficiency | Time per Iter (FB dataset) | GPU Memory-Usage (FB dataset) |
> |:---------------------------------------:|:-------------------:|:---------------:| :---: |:--------------------------:|:-----------------------------:|
> |             GCN (standard)              |        N.A.         |      N.A.       | N.A. |          0.2841s           |           414.8 MiB           |
> |  Lipschitz originally for CNN/MLP [1]   |     Restrictive     |       No        | No        |          3.2407s           |          2130.1 MiB           |
> | Layered Lipschitz Computing for GNN [2] |     Restrictive     |       Yes       | No        |          3.8541s          |          3280.7 MiB           |
> |          Ours (GCN + JacoLip)           |     Restrictive     |       Yes       | Yes       |          0.2972s           |           420.0 MiB           |
>
> ___
>
> **[W2. The proposed approach involves computing norms of the Jacobian matrix in the loss function. It can be memory and compute-intensive.]**
>
> A key to our approach is the mitigation of the memory and computational challenges commonly associated with computing the norms of the Jacobian matrix. We achieve this efficiency by avoiding the direct computation of the Jacobian matrix and consequently, the need for second-order derivative calculations.
>
> Specifically, our method capitalizes on the capability to compute the norm of the Jacobian matrix without the necessity to explicitly form the Jacobian itself. This approach eliminates the requirement for gradient of gradients computation. The key segment of our code (https://tinyurl.com/JacoLip, anonymized for review) is:
>
>
> ```python
> for i in range(out.shape[1]):
>     # Directional derivative vectors are initialized as zero vectors
>     v = torch.zeros_like(out)
>     v[:, i] = 1
>
>     # Compute gradients with respect to the inputs, not the model parameters
>     gradients = autograd.grad(outputs=out, inputs=input, grad_outputs=v,
>                               create_graph=True, retain_graph=True, only_inputs=True)[0]
>
>     # We only compute the norm of these gradients, which is a first-order operation
>     grad_norm = torch.norm(gradients, dim=1).unsqueeze(dim=1)
>     lip_mat.append(grad_norm)
> ```
>
> This code snippet demonstrates the computation of the gradient concerning the inputs, which is a first-order derivative. While we set the `create_graph=True` parameter, which allows for higher-order derivative computations, our method only utilizes this for first-order derivative calculations. The rationale is that our Lipschitz constant approximation demands only the norms of these gradients, not the gradients of these norms. Consequently, the computational complexity of our method remains at the first-order derivative level, with only a marginal increase compared to standard training procedures.

---

> ### Author Response · Authors · 2023-11-19
> **Response to Reviewer 7Erc (2/3)**
>
> Furthermore, we implement an efficient batching of gradient norm computations and do not retain the computational graph of these norms. This approach ensures that the memory footprint does not scale with the size of the Jacobian, but increases linearly with the output size. Given that the output size in relational datasets typically corresponds to a small dimension (i.e., the number of classes), the increase in memory requirement is minimal and manageable.
>
> Additionally, our method leverages the inherent efficiencies of PyTorch’s autograd system. This system dynamically allocates and deallocates memory during the training process, optimizing gradient computations and ensuring minimal memory usage by retaining only necessary gradients.
>
> In summary, our approach computes the Lipschitz constant without incurring the significant computational overhead associated with second-order derivatives. Our empirical results, supported by the provided code [https://tinyurl.com/JacoLip], affirm that our method does not result in substantial increases in training time or GPU memory usage when compared with standard training procedures.
>
> **This analysis is included in Appendix C.5 of our revised paper.**
>
> ___
>
> **[Q1. What intermediate tensors are being referred to here? ("... computational approach utilizes intermediate tensors extracted..." in Introduction)]**
>
> In the context of our work, the term "intermediate tensors" refers to the tensors that hold the computed gradients of the output with respect to the inputs of the model. These gradients are crucial for the computation of the Lipschitz constant but do not represent the final output or loss by themselves. Instead, they serve as a step in the process of evaluating the Lipschitz constant, which is then used to regularize the training. To illustrate, we quote the following segment of our implementation:
>
>
>
> ```python
> gradients = autograd.grad(outputs=out, inputs=input, grad_outputs=v, create_graph=True, retain_graph=True, only_inputs=True)[0]
> ```
>
> Here, `gradients` is a direct outcome of the computational graph’s backward pass. It contains the first-order derivatives of the output (out) with respect to the input (input). These gradients are used to compute the norm for each dimension of the output, which are then aggregated to estimate the Lipschitz constant of the model. Specifically, the process involves:
>
> 1. Computing directional derivatives (‘gradients’) of the model’s output with respect to its input.
> 2. Calculating the norm of these gradients for each output dimension.
> 3. Aggregating the norms to compute the Lipschitz constant.
> The efficiency of our computational strategy stems from the fact that we directly calculate the norms from these intermediate tensors, avoiding the explicit construction and storage of the full Jacobian matrix, which would be computationally prohibitive for large-scale graph datasets.
> In summary, the “intermediate tensors” refer to the tensors of gradients calculated during implicit backpropagation of forward, which are subsequently used to compute the Lipschitz constant in an efficient and scalable manner.
>
> **We make sure this is clarified in the revision's introduction section.**
>
> ___

---

> ### Author Response · Authors · 2023-11-19
> **Response to Reviewer 7Erc (3/3)**
>
> ___
> **[Q2. Eq 7 describes the Jacobian of the i-th node. Shouldn't it be a 3D tensor? i.e., $Y_{jk}$ (output of node_j) depends on $X_i$ (node_i) due to message passing. Are $\frac{\partial Y_{j1}}{\partial X_{i1}}$ for $i \neq j$ and such terms ignored? The discussion about the computation of the Lipschitz constant could be more elaborate, with a clear indication of the dimension of the tensors.]**
>
> Thank you for your insightful question. We would like to clarify that in our framework, the Jacobian matrix is indeed a 2D tensor. We specifically does not account for cases where $i \neq j$:
>
> 1. The term $\frac{\partial Y_{j1}}{\partial X_{i1}}$ where $i \neq j$ pertains to a global Lipschitz constant, considering the Lipschitz relationships for all nodes. In contrast, our paper focuses on local Lipschitz constants (specifically $\frac{\partial Y_{i1}}{\partial X_{i1}}$). As discussed in Section 3 of paper [1], computing global Lipschitz constants for a 2-layer MLP is NP-Hard. Therefore, extending this to even simple models like GCN and SGC would dramatically increase computational time, making it impractical.
>
> 2. However, we argue that global Lipschitz constants are not congruent with the principle of graph individual fairness, which is a cornerstone of our work. In our approach, we strive to maintain the similarity of like nodes from the input space to the output space, a process integral to ensuring individual fairness. This is achieved by controlling the expansion of data from the input to the output space, thereby reducing the effect of variations in input data on the output, a challenge we address through Lipschitz constraints.
>
>     When considering global Lipschitz constants, the scenario where $i \neq j$, the constant reflects the relationships of the entire relational network, encompassing each node's interaction with every other node. This broad perspective does not specifically account for minor perturbations or the nuances of local interactions. Instead, it generalizes the relationships across diverse nodes, considering the influence of all nodes on each individual node. This approach leads to a complex computational process, as it requires accounting for the interactions between all pairs of nodes across the network.
>
>     On the other hand, our methodology aligns more closely with local measures of fairness, as indicated in the paper [4], which emphasizes the importance of nearest-neighbor relationships. This local similarity measure, based on the nearest k neighbors, is more in line with our use of local Lipschitz constants. Our local approach focuses on the immediate vicinity of each node, ensuring that nodes are influenced primarily by their direct connections. This is not only computationally more manageable but also respects the inherent local structure of graph data.
>
>     Therefore, while global Lipschitz constants provide a comprehensive view of the network's dynamics, they fall short of addressing our objective of preserving individual fairness at the local level. Our preference for local Lipschitz constants is specifically tailored to maintain and respect the local relational dynamics within graph networks, ensuring each node is fairly represented based on its immediate connections.
>
>
> 3. In this context, when $i=j$, the **local** Lipschitz constants are more aligned with the objective, as there is no need to maintain pairwise similarity between all nodes in the output space: a node is solely influenced by its neighboring nodes. In cases where nodes exhibit a high degree of similarity, the variation in features between any two nodes can be understood as minor perturbations. Our focus, therefore, is on assessing how these small changes impact the output, which is effectively measured using gradient calculations. In situations where the gradient is particularly steep, even slight perturbations can have a substantial effect on the output.
>
> The Jacobian matrix is an effective tool for determining the Lipschitz constants for each node and its dimensions. Computational frameworks like PyTorch facilitate the acceleration of this computation process. This approach reduces the overall algorithmic complexity, enhancing the adaptability of our method while simultaneously ensuring alignment with the principles of individual fairness,
>
> **We integrate this update into Appendix B.4 of our revised paper. Additionally, in Appendix B.5 we discuss the Lipschitz constant (first mentioned in Preliminaries Section) and its relation to individual fairness for straightforward clarity.**
>
> ___
>
> **References:**
>
> [1] Efficiently Computing Local Lipschitz Constants of Neural Networks via Bound Propagation, NeurIPS 2022
>
> [2] Enhancing Node-Level Adversarial Defenses by Lipschitz Regularization of Graph Neural Networks, KDD 2023
>
> [3] Spectral Norm Regularization for Improving the Generalizability of Deep Learning
>
> [4] Individual Fairness For Graph Neural Networks: A Ranking Based Approach, KDD 2021

---

> > ### Comment · Reviewer_7Erc · 2023-12-03
> > **Thanks for the response**
> >
> > Thanks for the elaborate response. My concerns have been addressed, and I will maintain my score.
> >
> > Minor suggestion: Add a pointer to App B.4 in the main paper (Sec 3, preferably) to inform the readers.

---

> ### Author Response · Authors · 2023-11-22
>
> We hope that our response has helped explain our work's contributions. Please feel free to let us know if you have any further questions.

---

### Meta-Review · Area_Chair_qEiw · 2023-12-06

**Metareview:**

The paper presents a new approach to upper bound the Lipschitz constant of graph neural networks. The main technique is to use Jacobian matrix as a proxy to estimate the variation of the network in terms of change in inputs. The paper also discusses how the approach can be useful to guarantee rank-based individual fairness. Experimental results are presented to demonstrate the usefulness of the algorithm.

**Strengths**
- The paper is clearly written and the approach is easy to follow. During the author rebuttal phase, part of the source code was also shared by the authors to address certain concerns.
- Though precise estimation on the Lipschitz constant is hard, this paper makes a step to find some simple proxy to give an upper bound, and evaluate the performance of their approach via extensive experiments.
- The Lipschitz constant estimation scheme is further integrated into a rank-based individual fairness problem.

**Weaknesses**
- No theoretical analysis on time complexity or memory cost is presented.
- The experiments can be better organized.
- The method may lead to loose estimate on the Lipschitz constant.

**Suggestions to authors**
- Authors are suggested to incorporate their response to reviewers into the revision.
- While it is uneasy to show the computational complexity for the full pipeline, authors are suggested to provide bounds on some key steps.

**Justification For Why Not Higher Score:**

The paper lacks theoretical justification on the running time. The theory part, while useful, is not insightful enough.

**Justification For Why Not Lower Score:**

The paper provides a useful approach for graph neural network. The problem is important, and the approach is simple and works well on a few real-world data sets.

---

### Decision · Program_Chairs · 2024-01-16

Accept (poster)